# DNA-Stimulated Liquid-Liquid phase separation by eukaryotic topoisomerase ii modulates catalytic function

Joshua Jeong[1], Joyce H Lee[1], Claudia C Carcamo[1], Matthew W Parker[2], James M Berger[1]*

[1]Department of Biophysics and Biophysical Chemistry, Johns Hopkins University School of Medicine, Baltimore, United States; [2]Department of Biophysics, University of Texas Southwestern Medical Center, Dallas, United States

**Abstract** Type II topoisomerases modulate chromosome supercoiling, condensation, and catenation by moving one double-stranded DNA segment through a transient break in a second duplex. How DNA strands are chosen and selectively passed to yield appropriate topological outcomes – for example, decatenation vs. catenation – is poorly understood. Here, we show that at physiological enzyme concentrations, eukaryotic type IIA topoisomerases (topo IIs) readily coalesce into condensed bodies. DNA stimulates condensation and fluidizes these assemblies to impart liquid-like behavior. Condensation induces both budding yeast and human topo IIs to switch from DNA unlinking to active DNA catenation, and depends on an unstructured C-terminal region, the loss of which leads to high levels of knotting and reduced catenation. Our findings establish that local protein concentration and phase separation can regulate how topo II creates or dissolves DNA links, behaviors that can account for the varied roles of the enzyme in supporting transcription, replication, and chromosome compaction.

## Editor's evaluation

Type II topoisomerases are essential players in virtually every aspect of genome organization and function of all organisms. The in vitro data presented here clearly demonstrate that eukaryotic type II topoisomerases phase separate under physiological conditions, forming liquid-liquid condensates, and that the outcomes of type topoisomerase II activity on DNA are altered in these condensates. The experiments and methods are sound, clearly described, and fully support the insightful and carefully formulated interpretation of the data. This work has broad implications for dissecting and delineating the myriad fundamental roles of this centrally important molecule.

*For correspondence:
jmberger@jhmi.edu

## Introduction

Type II topoisomerases resolve DNA supercoiling and entanglements that arise naturally from the action of transcription, replication, and repair machineries on chromosomes (**Chen et al., 2013**). Enzymes of this family modulate DNA topology by an ATP-dependent strand passage mechanism in which one duplex segment is physically passed through a transient, protein-mediated break in a second double-stranded DNA (**Brown and Cozzarelli, 1979**; **Wang, 1998**). Two classes of type II topoisomerases have been identified to date: type IIA, which comprise proteins such as eukaryotic topo II, prokaryotic topo IV and DNA gyrase, and type IIB, which include two isoforms, topo VI and topo VIII, which are found in plants, archaeal organisms, and some viruses (**Forterre et al., 2007**).

**Figure 1.** Topo IIs alter DNA topology. (*Left*) Schematic of how topo IIs transform DNA from one topological state to another. (*Right*) Depiction of how different DNA topological states migrate on a native-agarose gel after electrophoresis. Colored arrows correlate with colored labels for different topological species.

Due to its double-helical nature, a closed DNA circle or anchored DNA loop can adopt several distinct topological states, such as becoming over- or under-wound to form a supercoiled species, becoming self-entangled to form a knot, or becoming linked with another DNA loop to form a catenane (*Figure 1*). Resolution of these structures requires DNA strand breakage and the physical movement of DNA segments through one another. Apart from gyrase, which adds negative supercoils to DNA (*Gellert et al., 1976*), all other type II topoisomerases tend to 'simplify' DNA topology – for example, relaxing supercoiled DNA, untying DNA knots, or unlinking catenated loops – in accord with a lowering of the torsional or entropic energy of the system (*Rybenkov et al., 1997*). However, type II topoisomerases can also actively form complicated DNA structures such as knots and catenanes, depending on conditions. For example, knotted DNAs can be formed in vitro using high concentrations of bacteriophage T4 topo II (*Wasserman and Cozzarelli, 1991*). Catenated DNAs can be generated by type II topoisomerases through the addition of proteinaceous or chemical DNA condensing agents (*Krasnow and Cozzarelli, 1982*; *Riou et al., 1985*; *Tse et al., 1984*). Both DNA knots and catenanes can be created in vitro when high concentrations of eukaryotic topo II are present relative to DNA (*Hsieh, 1983*).

Overall, the relative concentrations of both DNA and enzyme appear to play a significant role in controlling whether type II topoisomerases selectively remove linkages (simplifying DNA topology) or add linkages (complicating topology). In cells, DNA is maintained at a high concentration (millimolar

base pairs) compared to that of the topoisomerase (e.g., human topo II is present at low-to-sub micro-molar levels in cells [*Padget et al., 2000*]), yet the enzyme is capable of altering its catalytic output to suit specific needs. For example, during transcription and replication, positive DNA supercoils generated by RNA polymerases or replisomes are recognized and relaxed by type II topoisomerases to prevent stalling (*Liu and Wang, 1987*; *Lockshon and Morris, 1983*; *Pruss, 1985*). Similarly, following replication, interlinked chromosomes are decatenated by enzymes like topo II and topo IV to prevent DNA breaks (*Cebrián et al., 2015*; *Hiasa and Marians, 1996*; *Lucas et al., 2001*; *Peter et al., 1998*). However, some type II topoisomerases can also create knots, such as the action of topo II in regions with high transcriptional activity (*Valdés et al., 2018*), a function that can locally compact DNA. Topo II has also been proposed to actively catenate DNA as a means of promoting chromosome condensation and/or cohesion (*Anderson and Roberge, 1996*; *Bauer et al., 2012*; *Farr et al., 2014*; *Ishida et al., 1994*; *Tavormina et al., 2002*). How type II topoisomerases selectively toggle between DNA linking *vs.* unlinking depending on chromosomal context has been a long-standing question in the field.

One clue to understanding the disparate topological outcomes mediated by type II topoisomerases can be found in vertebrates, which express two different topo II isoforms – Top2α and Top2β – that have specific cellular roles. Top2α is essential in dividing cells where it assists in cell-cycle functions such as chromosome condensation and segregation (*Gonzalez et al., 2011*; *Johnson et al., 2009*; *Nielsen et al., 2020*; *Samejima et al., 2012*; *Spence et al., 2007*). By comparison, Top2β plays a role in supporting transcriptional initiation (*Edmond et al., 2017*; *Haffner et al., 2010*; *Lyu et al., 2006*). Evidence suggests that functional differences between Top2α and Top2β derive in part from a variable, intrinsically disordered C-terminal domain (CTD) appended to an evolutionarily conserved catalytic core (*Linka et al., 2007*). The CTD is subject to extensive post-translational modifications (*Antoniou-Kourounioti et al., 2019*; *Bedez et al., 2018*; *Cardenas et al., 1992*; *Grozav et al., 2011*; *Mao et al., 2000*), has been shown to mediate interactions with different partner proteins (*Lane et al., 2013*; *Nakano et al., 1996*; *Niimi et al., 2001*; *Yamane et al., 1997*) and, specifically for Top2α, appears to help localize the enzyme to specialized chromosomal regions such as centromeres (*Andersen et al., 2002*; *Pandey et al., 2020*; *Yoshida et al., 2016*). How the topo II CTD manifests its varied activities has yet to be established. Interestingly, some proteins known to associate with topo IIs and some genetic loci where the enzyme is enriched have been proposed to undergo liquid-liquid phase separation (LLPS), forming assemblies that distinguish them from general chromatin areas (*Boija et al., 2018*; *Cebrián et al., 2015*; *Cho et al., 2018*; *Frattini et al., 2021*; *Trivedi et al., 2019*; *Yamane et al., 1997*; *Zamudio et al., 2019*). During LLPS, macromolecules locally concentrate to form a new fluid phase that demixes from the bulk solution to establish a dynamic, membraneless compartment (*Banani et al., 2017*; *Shin and Brangwynne, 2017*). Whether and how topo II might interface and cluster with such elements is unknown.

Here, we report that budding yeast topo II (*Sc*Top2) and both human TOP2α (*Hs*TOP2α) and TOP2β (*Hs*TOP2β) can phase separate to form condensed bodies. Condensation occurs at physiological protein concentrations and is stimulated by DNA, which also fluidizes the condensates. The CTD of *Sc*Top2 is shown to be both necessary and sufficient to drive condensation in the presence of DNA, and its phosphorylation status impacts the fluidity of condensates formed with the full-length enzyme and DNA. Unexpectedly, condensation is found to markedly change the catalytic output of not only *Sc*Top2 but also both human topo IIs, shifting from the simplification of topological complexity (e.g. supercoil relaxation) to the robust formation of more complex DNA structures (catenanes); by contrast, removal of the topo II CTD in *Sc*Top2 not only prevents LLPS but also shifts enzyme output primarily to the production of DNA knots at enzyme concentrations where condensates would otherwise form. Collectively, our findings define a molecular role for the enigmatic CTD of topo II, showing that the enzyme uses this element to promote inter-topoisomerase interactions, locally concentrate distal DNA segments, and modulate topological complexity. These properties help explain how the strand passage mechanism of topo II can be selectively controlled to support chromosomal transactions that occur in specific chromosomal contexts and require distinct topological outcomes.

# Results

## Budding yeast topo II forms higher-order collectives in solution

Limited proteolysis studies (*Austin et al., 1995*), as well as primary sequence analysis using the disordered protein prediction algorithm IUPRed2A (*Erdős and Dosztányi, 2020*), indicate that the CTDs of eukaryotic topo IIs are long, intrinsically disordered regions (IDRs) (*Figure 2A*). IDRs have been shown to facilitate the self-assembly of macromolecules into liquid-like condensates through phase separation in a variety of protein systems (*Banani et al., 2017*; *Lyon et al., 2021*). Many proteins that undergo phase separation possess IDRs that exhibit low amino acid sequence complexity (*Franzmann and Alberti, 2019*; *Uversky, 2017*); exceptions include factors that support the initiation of DNA replication in metazoans, which have an IDR sequence complexity similar to that of folded proteins except for a lack of aromatic residues (*Parker et al., 2021*; *Parker et al., 2019*), and the Nephrin Intracellular Domain (NICD) (*Pak et al., 2016*). Analysis of the relative amino-acid composition of topo II CTDs shows that despite having relatively sparse sequence homology to each other (*Figure 2—figure supplement 1*), the elements generally have a high lysine content (≥15%), a moderately high amount of serine (~10%), aspartate, and glutamate (10%), and a very low percentage of each aromatic amino acid (<5%) (*Figure 2*). Despite these similarities, eukaryotic topo II CTDs can have different compositions of positively and negatively charged amino acids and varied isoelectric points (pI = 5.5–9.6) (*Figure 2B*). Overall, the compositional properties of the topo II CTDs are more similar to those of the eukaryotic replication initiators, as opposed to low-complexity sequences found in representative LLPS-forming proteins such as human FUS (*Wang et al., 2018a*) or DDX4 (*Nott et al., 2015*; *Figure 2—figure supplement 2*).

Since the IDRs of eukaryotic topo IIs share compositional similarities with eukaryotic replication initiators, which undergo phase separation (*Parker et al., 2019*), we asked whether they could also form condensates in vitro. To address this question, we first focused on budding yeast topo II (*Sc*Top2). We purified *Sc*Top2, N-terminally labeled the protein with a Cy3 fluorophore, buffer exchanged the sample into high salt (600mM), and diluted the labeled protein to physiological salt concentrations (150mM) to induce phase separation. Potassium acetate (KOAc) was chosen as a salt for these and other subsequent experiments, as $K^+$ is the predominant monovalent cation and acetate is a high abundance anion in both human and yeast cells (internal $Na^+$ and $Cl^-$ concentrations are negligible in these systems) (*Ince et al., 1987*; *Rimareva et al., 2017*). After dilution, samples were imaged using fluorescence confocal microscopy (*Figure 2C*). At the lowest concentration of *Sc*Top2 tested (50nM), no visible bodies were seen. However, at moderate protein concentrations (375nM), small circular fluorescent bodies became apparent. These bodies (referred to hereafter as 'puncta') grew in size and number as protein concentration increased. While conducting these experiments, we noted that the formation of puncta was time-dependent, with puncta appearing within 10min of mixing and plateauing in number and size after~60min (*Figure 2—figure supplement 3A*). To ensure we were operating under a relatively equilibrated regime, the subsequent experiments were analyzed 1.5hr after condensation was induced.

Because topo IIs are DNA-binding proteins, we next assessed whether the observed puncta could incorporate DNA. Cy3-*Sc*Top2 was mixed with a Cy5-labeled 200bp DNA duplex and samples were again imaged by confocal microscopy. Imaging at Cy3 and Cy5 excitation wavelengths showed that both protein and DNA co-localize in the puncta; the labeled DNA on its own did not form such bodies (*Figure 2D*). DNA-dependent stimulation of puncta formation was also observed when Cy3-*Sc*Top2 was mixed with an unlabeled negatively supercoiled 2.9kb plasmid (pSG483) (*Figure 2D*). Interestingly, puncta formed in the presence of DNA were noticeably larger compared to those formed without (*Figure 2—figure supplement 3B*). Collectively, these data show that budding yeast topo II can form higher-order spherical assemblies at protein concentrations greater than 250nM and that DNA both stimulates the formation of and is integrated into these bodies.

To further probe the mechanism by which DNA promotes condensation of topo II, we next conducted a concentration- and length-dependent titration of DNA against 500nM Cy3-*Sc*Top2 and imaged the samples by fluorescence confocal microscopy (*Figure 2E*, *Figure 2—figure supplement 4*). The shortest DNA tested (50bp, roughly the binding site size of the catalytic region) did not increase the number nor the size of condensates at any concentration tested. By comparison, the 100bp, 200bp, and pSG483 DNA substrates all enhanced condensate formation at intermediate mass concentrations (5–20ng/μL). To determine whether the stimulation of condensation was dependent

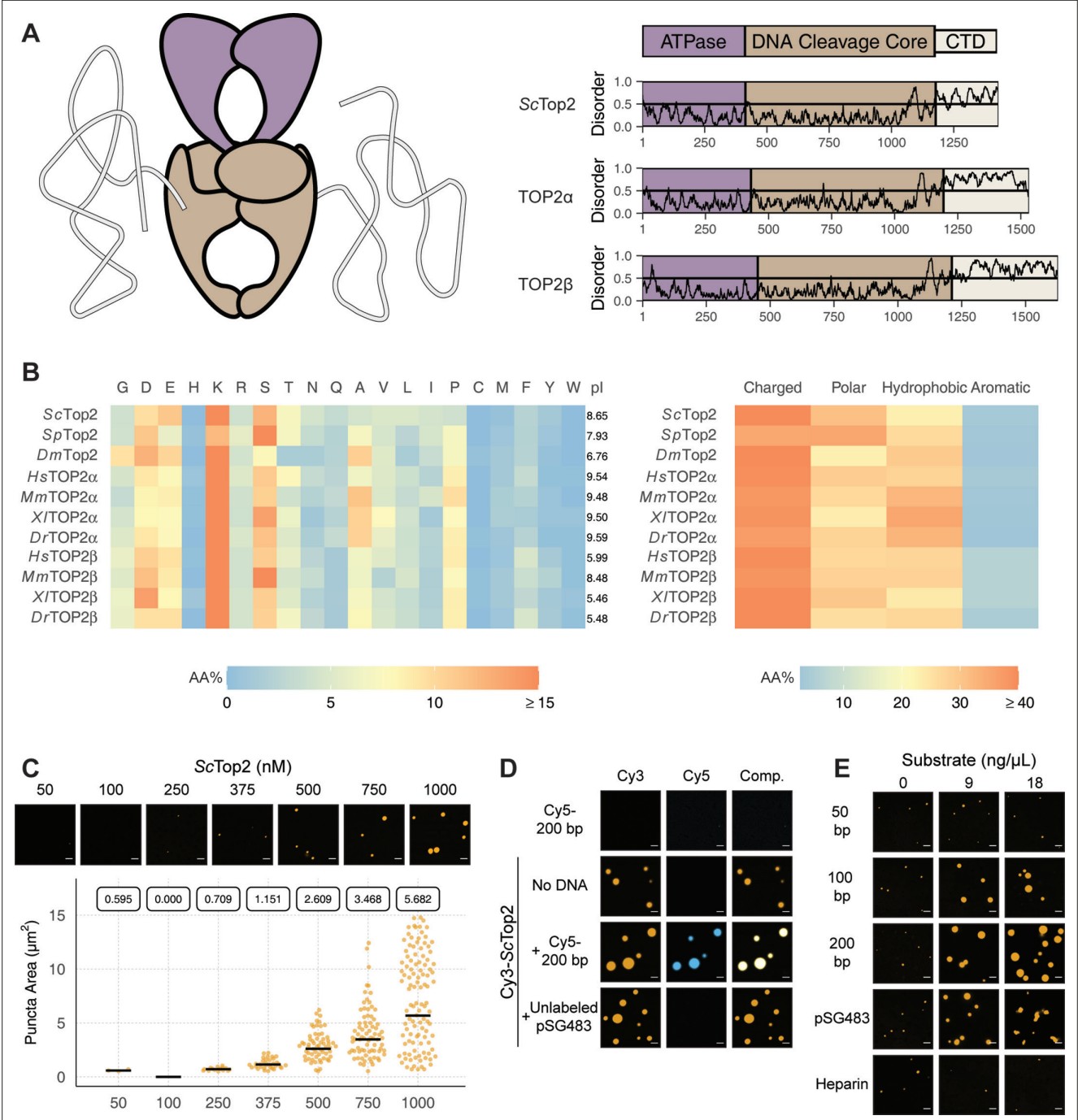

**Figure 2.** *Eukaryotic topo IIs possess long IDRs and form higher order assemblies.* (**A**) General schematic of eukaryotic topo II with color coded domains on right. Graphs show disorder tendency of primary sequence calculated by IUPred2A for budding yeast and human topo IIs. Values above 0.5 signify disorder. (**B**) Relative amino acid composition of various eukaryotic topo II CTDs with respective isoelectric points and relative breakdowns of amino acid composition and groups. (**C**) Representative images of *Sc*Top2 at different concentrations under phase-separating conditions. The quasirandom plot quantifies puncta number across three different micrographs for each condition. The solid bar bisecting the plots corresponds to the geometric mean area for each set of data (quantified by the boxed number). (**D, E**) Representative images of (**D**) 1µM Cy3-*Sc*Top2 and (**E**) 500nM Cy3-*Sc*Top2 with and without DNA substrates or, in (**E**), with heparin. The concentration of Cy5-200 bp DNA and pSG483 in (**D**) is 50nM and 5nM, respectively. 50nM Cy5-200 bp in the absence of protein is shown as a negative control. Scale bar in (**C**), (**D**), and (**E**) is 5µm.

The online version of this article includes the following figure supplement(s) for figure 2:

**Figure supplement 1.** Topo II C-terminal domains have sparse sequence homology.

*Figure 2 continued on next page*

*Figure 2 continued*

**Figure supplement 2.** Quantification of relative amino acid percentages of IDRs that undergo LLPS in vitro.

**Figure supplement 3.** Quantification and distribution of puncta formation across 120min for *Sc*Top2 in the (**A**) absence and (**B**) presence of DNA.

**Figure supplement 4.** Quantification and distribution of puncta present as a consequence of DNA or heparin addition.

on DNA, a titration was also conducted with another polyanionic substrate, heparin. Interestingly, heparin failed to stimulate DNA condensation at the observed concentrations. Together, these findings establish that condensate formation by *Sc*Top2 is aided by extended DNA segments and is not stimulated simply by the presence of a poly-anionic substrate.

## ScTop2-DNA assemblies possess fluid-like characteristics

We next sought to determine whether puncta formed by *Sc*Top2 display liquid-like properties. Nucleic-acid binding proteins that undergo phase separation tend to exhibit a biphasic response with respect to the relative concentration of DNA or RNA (*Du and Chen, 2018*; *Maharana et al., 2018*; *Zhang et al., 2015*): as a nucleic acid substrate is titrated against a fixed amount of protein, condensation is first promoted, but then as concentrations are increased further, the protein is solubilized and condensation is antagonized. To determine whether puncta formed by *Sc*Top2 with DNA behave in this manner, varying concentrations of unlabeled pSG483 were mixed with different amounts of the Cy3-labeled protein (*Figure 3A*, *Figure 3—figure supplement 1*). At low DNA concentrations (5–10nM), the number and size of puncta increased relative to samples in the absence of DNA. By contrast, as the DNA concentration was increased further, puncta disappeared. Interestingly, at the intermediate concentrations of DNA tested (25 and 50nM), observed bodies took on elongated, fibril-like shapes for the plasmid substrate (*Figure 2E*); by comparison, puncta formed with short DNAs

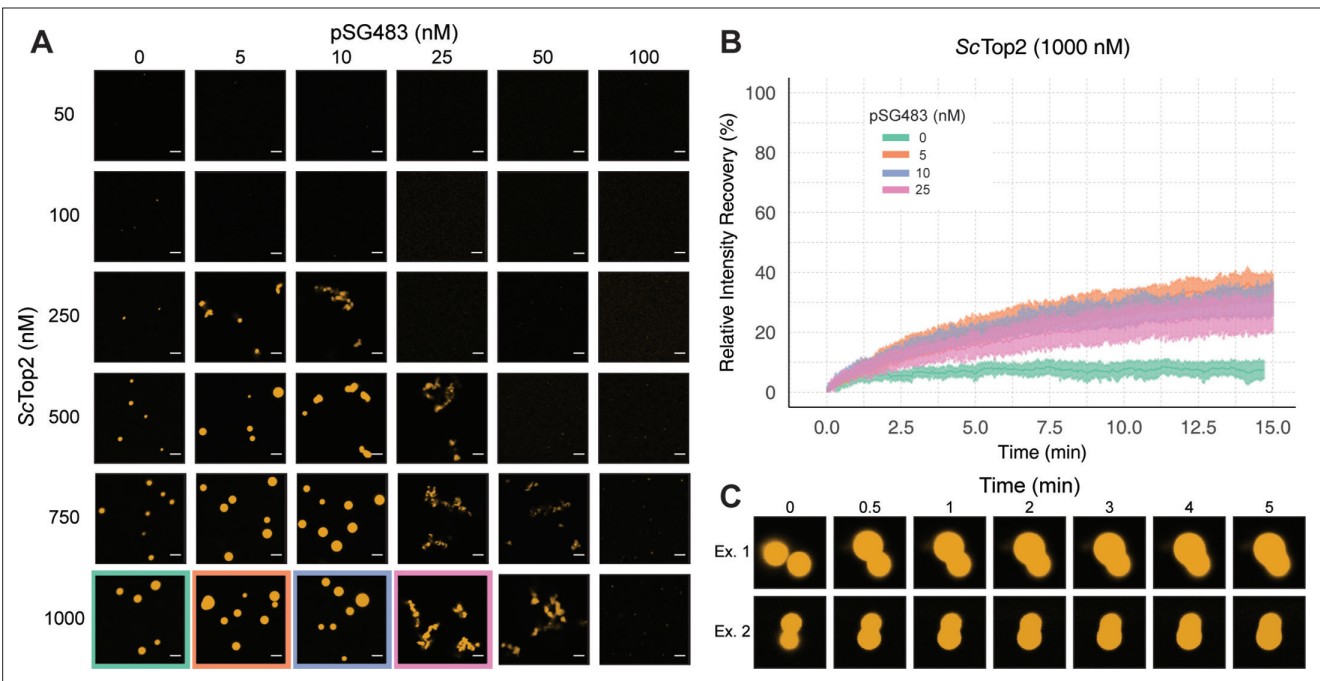

**Figure 3.** Sc*Top2 higher order assemblies possess liquid-like qualities in the presence of DNA.* (**A**) Representative images of assemblies formed during pairwise Cy3-*Sc*Top2 and pSG483 concentration titrations. (**B**) Quantification of relative intensity recovered from puncta over 15min following photobleaching. For each condition, 10 different puncta were bleached across three different technical replicates. Data are fit to a mean-of-relative-intensity curve and shaded regions are the standard deviation of each sample. (**C**) Representative images of 1μM Cy3-*Sc*Top2 and 5nM pSG843 time-lapses at specific time points to show fusion events between puncta. Scale bar in (**A**) is 5μm.

The online version of this article includes the following figure supplement(s) for figure 3:

**Figure supplement 1.** Quantification and distribution of puncta formation in response to increasing amounts of pSG483.

remained spherical. This size-dependent type of behavior has been previously seen with other DNA-binding proteins that undergo LLPS, such as HP1α (*Keenen et al., 2021*).

One key attribute of true liquid-like (as opposed to gel-like) condensates is that molecules can freely diffuse inside the droplets, which themselves have an ability to coalesce into larger condensates. To test whether *Sc*Top2 can freely diffuse inside the bodies seen in our micrographs, puncta were formed in the absence or presence of increasing concentrations of pSG483 and subjected to fluorescence recovery after photobleaching (FRAP) (*Figure 3B*). In reactions lacking DNA, the intensity of the Cy3-*Sc*Top2 puncta did not recover following photobleaching, suggesting that these bodies possess more gel-like characteristics (*Frey et al., 2006*; *Jawerth et al., 2020*). By contrast, in the presence of 5nM unlabeled pSG483, a fraction of the Cy3-*Sc*Top2 intensity (roughly one-third) had recovered by the same 15minute mark. As the pSG483 concentration increased, the mobile protein fraction inside the puncta appeared to decrease, with the fibrillar bodies (25–50nM DNA) showing somewhat less fluidity than the spherical droplets (5–10nM DNA). Time-lapse microscopy of puncta containing *Sc*Top2 and 5nM pSG483 also revealed fusion events between puncta on the order of minutes (*Figure 3C*). Thus, while the smaller puncta formed by ScTop2 on its own are relatively static, the protein in both the spherical and fibrillar structures formed by *Sc*Top2 in the presence of DNA is dynamic, exhibiting the defining attributes of condensates formed by LLPS.

## Ionic coacervation by the CTD constitutes the multivalent interactions that drive LLPS by ScTop2

We next sought to determine whether the CTD of topo II, owing to its intrinsically disordered nature, is responsible for supporting LLPS. We purified both the *S. cerevisiae* enzyme lacking this region (*Sc*Top2^ΔCTD) and the *Sc*Top2 CTD itself. Both proteins were N-terminally labeled with Cy3 and prepared in the same manner as wild type *Sc*Top2 before imaging with confocal microscopy (*Figure 4A*). The Cy3-ScTop2^ΔCTD construct did not form condensates in either the presence or absence of a 200bp Cy5-labeled DNA, even at protein concentrations as high as 1μM. By comparison, the Cy3-labeled CTD failed to form condensates in the absence of DNA but did form bodies in the presence of 50nM Cy5-labeled 200bp duplex at a lower salt concentration of 50mM KOAc (*Figure 4A*). Interestingly, several puncta formed with the CTD alone integrated DNA into only a portion of the bodies, as compared to the fully homogenous distribution of DNA seen with full-length *Sc*Top2 (*Figure 4A*). The addition of PEG as a crowding agent also induced puncta formation by the CTD, but in a manner that did not require DNA, indicating that this region can weakly associate with itself (*Figure 4—figure supplement 1*). Overall, these findings demonstrate that the *Sc*Top2 CTD is necessary for supporting LLPS by the full-length enzyme and that DNA can act as a scaffold to promote co-localization of the element.

The enrichment of charged amino acids in the *Sc*Top2 CTD suggested that ionic interactions might play a particularly key role in facilitating LLPS. To test this assumption, 500nM Cy3-*Sc*Top2 was titrated with varying concentrations of pSG483 across four different salt concentrations (*Figure 4B*, *Figure 4—figure supplement 2*). In the absence of DNA, the small puncta formed by *Sc*Top2 alone at the control potassium acetate concentration (150mM) disappeared as soon as salt levels were increased to 200mM. By comparison, in the presence of pSG483, the large condensates seen at 150mM salt persisted until potassium acetate levels reached 400mM. Interestingly, the intermediate salt concentrations still supported solubilization of *Sc*Top2 condensates by elevated levels of plasmid DNA. To determine whether aromatic contacts might contribute to promoting phase separation by *Sc*Top2, we added 1,6-hexanediol (1,6-HD), an aliphatic alcohol frequently used to disrupt such interactions, to condensation reactions (*Figure 4B*, *Figure 4—figure supplement 2*). Interestingly, 1,6-HD strongly promoted the aggregation of *Sc*Top2 into large, irregular bodies in both the presence and absence of DNA while also preventing the resolubilization of the bodies when pSG483 concentrations were increased. The addition of higher levels of 1,6-HD (e.g. 10%) also severely disrupted DNA supercoil relaxation by *Sc*Top2 (*Figure 4—figure supplement 2*), indicating that the agent has a detrimental effect on the protein.

To isolate any effects that salt or 1,6-HD might have on the topo II catalytic core as compared to the CTD, we next assessed the action of both additives on the *Sc*Top2 CTD alone. As seen previously, the CTD was unable to form condensates in the absence of DNA, even when the salt concentration was lowered to 50mM potassium acetate (*Figure 4—figure supplement 2*). However, robust CTD-DNA interactions were seen at this lower salt concentration, as evidenced not only by the presence of

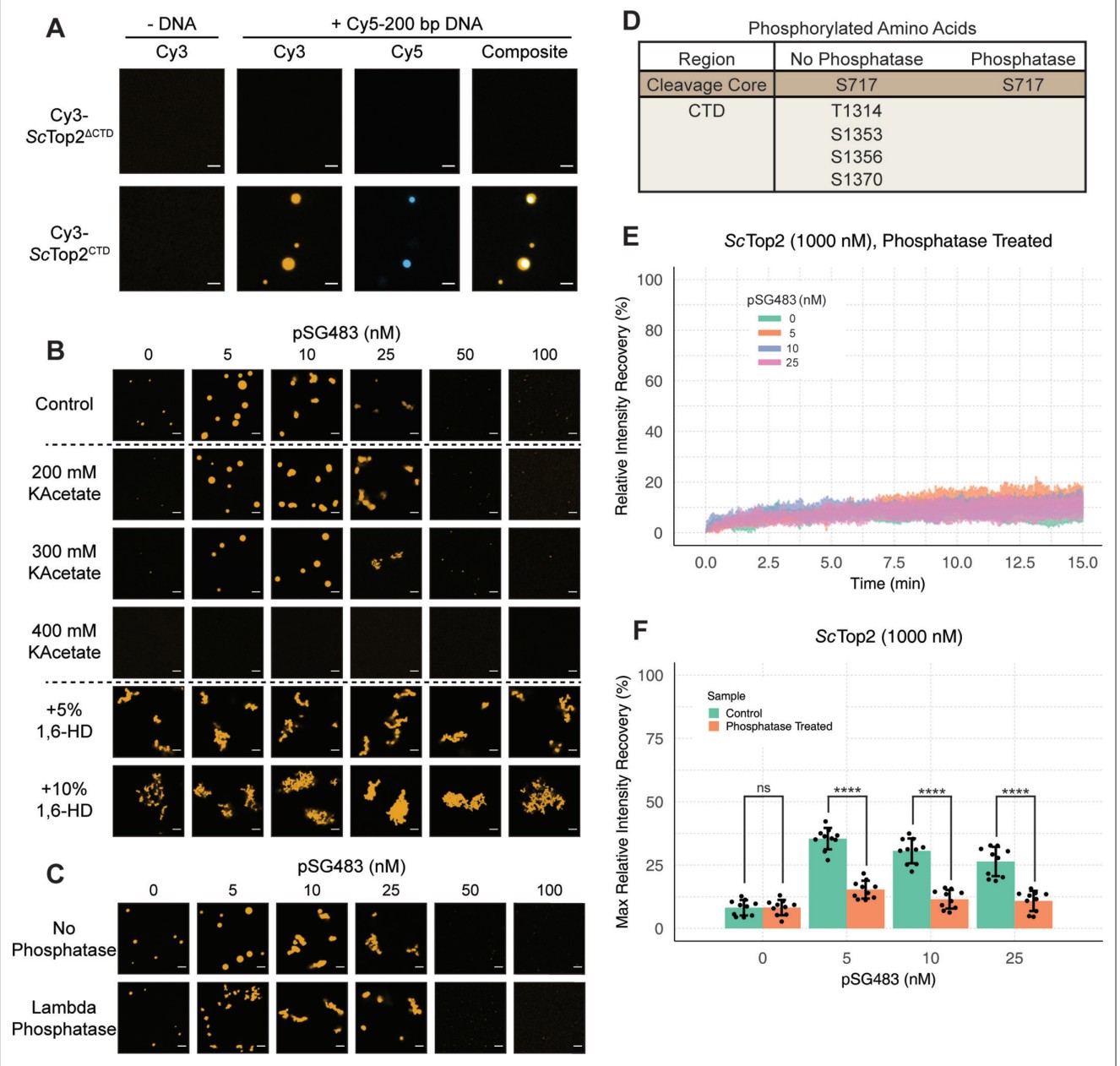

**Figure 4.** *Ionic, protein-DNA interactions drive LLPS by ScTop2.* (**A**) Representative images of 1μM Cy3-*Sc*Top2^ΔCTD in 150mM KOAc and 1μM *Sc*Top2^CTD in 50mM KOAc mixed with and without 50nM Cy5-200 bp DNA. (**B**) Representative images of 500nM Cy3-*Sc*Top2 mixed with varying concentrations of pSG483 in the presence of different buffer conditions. Control reaction has 150mM KAcetate. Salt concentrations represent the final values in solution. 1,6-hexanediol (1,6-HD) weight by volume percent was added to 150mM KAcetate solution. (**C**) Representative images of puncta/fibrils comparing phosphatase-treated and untreated *Sc*Top2. (**D**) List of phosphorylated amino acids as identified by mass spectrometry analysis. (**E**) FRAP analysis of phosphatase-treated protein. (**F**) Quantification of mobile phase for *Figures 2B and 3E* Error bar is standard deviation for each condition. Two-Way ANOVA with bonferroni post hoc analysis was done on ten puncta per condition. ns is non-significant, **** means p<0.0001. Scale bars in (**A**), (**B**), and (**C**) are 5μm.

The online version of this article includes the following figure supplement(s) for figure 4:

**Figure supplement 1.** *Sc*Top2 CTD forms puncta in the presence of PEG.

**Figure supplement 2.** Quantification and analysis of different additives on *Sc*Top2.

**Figure supplement 3.** Phosphorylation status of *Sc*Top2 alters condensate properties.

large numbers of large puncta, but also by the co-localization of DNA inside the CTD bodies; as salt concentration was increased, puncta became more diffuse. By comparison, 1,6-hexanediol had little effect on condensate formation by the CTD. Overall, our data indicate that condensate formation by *Sc*Top2 is primarily reliant on ionic protein-DNA interactions and, consistent with its Tyr-, Phe-, and Trp-poor amino acid composition (*Figure 2B*), not on contacts involving aromatic amino acids.

The propensity of IDRs to form condensates can be fine-tuned by post-translational modifications (*Larson et al., 2017*; *Nott et al., 2015*; *Ryan et al., 2018*; *Wang et al., 2018b*; *Wang et al., 2014*). Since topo II CTDs are modified by phosphorylation (*Bedez et al., 2018*; *Cardenas et al., 1992*), we asked whether phosphorylation status might impact the enzyme's ability to undergo LLPS. To do this, *Sc*Top2 was purified and treated with lambda phosphatase; the phosphatase and non-phosphatase treated proteins were then mixed with varying concentrations of pSG483 and imaged by fluorescence confocal microscopy (*Figure 4C*). In the absence of DNA, the phosphatase-treated and untreated (control) enzymes both had the same condensate-forming propensity. However, as pSG483 was first added (5nM), phosphatase treated *Sc*Top2 produced a larger number of small puncta as compared to the smaller number of large droplets formed with the control reaction (*Figure 4C*, *Figure 4—figure supplement 3*). As pSG483 concentration was further increased (25nM), the phosphatase-treated sample produced more elongated fibrillar bodies than the control. At still higher DNA levels, puncta were no longer apparent, indicating that solubilization had occurred. Mass spectrometry analysis confirmed that the phosphatase stripped phosphates off from all amino acids in the CTD (*Figure 4D*). Reasoning that the observed difference in behavior between native and phosphatase-treated ScTop2 at lower DNA concentrations might be due to differences in the fluidic properties of the bodies formed by the dephosphorylated protein, we conducted FRAP studies of the puncta present in the phosphatase-treated reactions (*Figure 4E*). Similarly to the native phosphorylated samples, phospha-tase- treated protein produced assemblies that failed to recover fluorescence intensity in the absence of DNA (*Figure 4F*). However, in contrast to the untreated protein, adding DNA did not increase the fluidity of the bodies formed by the phosphatase-treated protein. Thus, the phosphorylation status of *Sc*Top2 does not control the ability of the protein to undergo LLPS but does change the physical behavior of the bodies that are formed when DNA is present.

## ScTop2 LLPS promotes DNA catenation

Phase separation in cells not only influences protein localization but can also modulate enzymatic activity (*Banani et al., 2017*; *Lyon et al., 2021*). To determine whether condensate formation might directly alter topo II catalytic function, we performed DNA supercoil relaxation assays under condensing and non-condensing conditions. *Sc*Top2 was buffer exchanged and diluted into a buffer equivalent to that used for phase separation assays, but with the addition of 1mM magnesium acetate, which is required for activity (this concentration of $Mg^{2+}$ has a negligible impact on phase separation (*Figure 5—figure supplement 1A*)). Protein and a fixed amount of pSG483 (25nM) were mixed and incubated at ambient temperature for 1.5hr to provide time for condensates to form. ATP was next added to start the assay; 5min later, reactions were stopped by the addition of EDTA, SDS and proteinase K, and analyzed by native agarose-gel electrophoresis. Before the addition of ATP, samples with Cy3-labeled *Sc*Top2 were imaged to check for the presence of condensates in the buffers used for the assays. Two different salt concentrations and six different enzyme concentrations were assessed to correlate condensate formation with reaction efficiency and the types of products produced (*Figure 5A*). At 150mM KOAc and 100nM *Sc*Top2, no condensates were apparent in images but supercoil relaxation had progressed to transform~80% of the starting substrate. By contrast, as protein concentration was increased, condensates became visible and pSG483 converted to a product that failed to enter the wells of the gel. When the salt concentration was elevated to 400mM, no condensates were visible at any enzyme concentration. Moreover, under these conditions, only relaxed DNA topoisomers were observed and no well-shifting was seen.

The use of SDS and proteinase K as quenching agents disrupts any protein aggregates that might retard DNA mobility during native gel electrophoresis. As a result, the appearance of DNA that was unable to migrate into the gel under condensing conditions suggested that the topological state of the plasmid substrate had changed. Catenated DNA networks, such as those formed in the kinetoplasts of certain protists (*Laurent and Steinert, 1970*; *Marini et al., 1980*, *Renger and Wolstenholme, 1972*, *Riou and Delain, 1969*), represent one DNA species that is unable to be electrophoresed

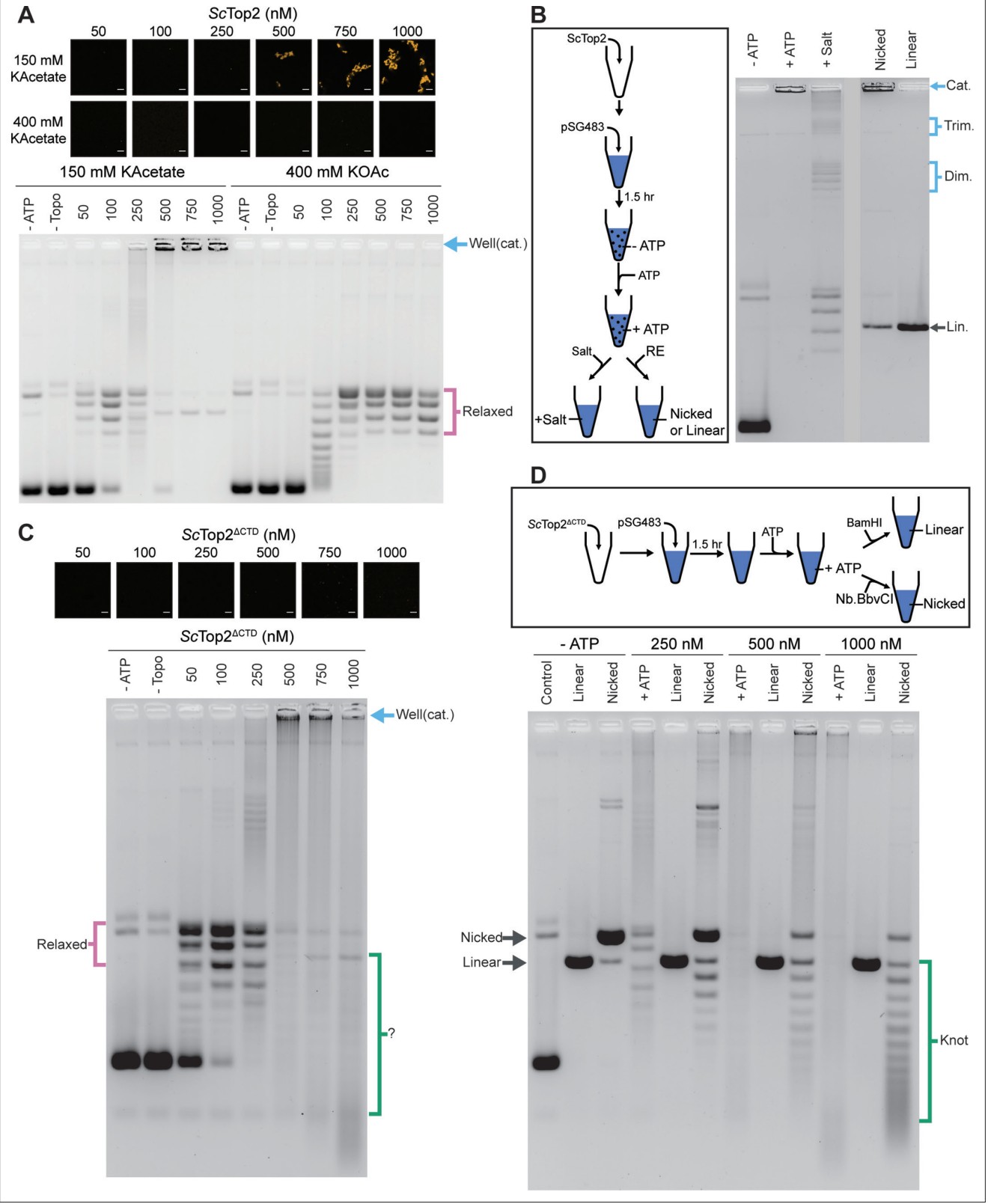

**Figure 5.** The ScTop2 CTD modulates protein catenation *versus* knotting. (**A**) Activity assay at different ScTop2 concentrations with 25nM pSG483 and either 150mM or 400mM KAcetate. LLPS does not occur at the higher salt concentration. Confocal images were taken before the addition of ATP and with Cy3-labeled protein. ScTop2 concentration in the 'no ATP' reaction is 1µM. No ATP and no topoisomerase reactions are shown as negative controls. (**B**) Modified activity assay with 1µM ScTop2 and 25nM pSG483. Workflow is shown at right. (**C**) Activity assay performed with ScTop2^ΔCTD and

*Figure 5 continued on next page*

*Figure 5 continued*

25nM pSG483. Confocal images were taken before the addition of ATP. Controls are similar to (**A**). (**D**) Modified activity assay of different *Sc*Top2$^{\Delta CTD}$ concentrations with 25nM pSG483 where reactions were treated with restriction enzymes to linearize or nick reaction products. '−ATP' samples show where supercoiled, linear, and nicked samples run on the gel. Workflow is shown at top. Scale bars in (**A**) and (**C**) are 5µm.

The online version of this article includes the following figure supplement(s) for figure 5:

**Figure supplement 1.** Magnesium effects and knotting propensity during LLPS by *Sc*Top2.

into an agarose gel matrix. To test whether the DNA products created by *Sc*Top2 under condensate-forming conditions are catenanes, we attempted to unlink the catenated network using topo II or restriction enzymes that either linearize or singly nick pSG483 (**Figure 5B**). The activity assay protocol was modified such that after the addition of ATP under condensing conditions, either: (1) the buffer was adjusted to a higher salt concentration (400mM) to dissolve the condensates and the reaction was quenched, or (2) restriction enzymes with a single cutting or nicking site in the plasmid were added directly to condensates (formed in 150mM KOac) to facilitate DNA linearization. Electrophoresis of the products revealed that the addition of salt results in the disappearance of the well-shifted species and the appearance of relaxed plasmid topoisomers, including what appear to be subsets of relaxed dimeric and trimeric catenanes. By comparison, treating the samples with a nicking enzyme resulted in the retention of most of the well-shifted species, while treating the samples with a restriction enzyme that linearizes pSG483 caused the disappearance of the well-shifted species and the appearance of a single band in the gel (**Figure 5B**). Collectively, these observations establish that the well-shifted DNA species formed by *Sc*Top2 in condensates is a network of interlinked (catenated) plasmids and that the catenation and decatenation activities of the enzyme can be directly controlled by phase separation.

To further explore the switch in activity seen for *Sc*Top2, assays were re-run at low (150mM) salt but now using the CTD-less construct, *Sc*Top2$^{\Delta CTD}$, which does not phase separate (**Figure 5C**). Nearly complete supercoil relaxation was observed at 100nM of this enzyme but now as protein levels were increased, extensive DNA streaking was seen, forming not just high but also low molecular-weight products, particularly at the highest enzyme concentration tested. Previous studies of the topological products formed by DNA topoisomerases suggested that some of these products might correspond to knotted DNAs. To test this hypothesis, the products of reactions using 250nM, 500nM, and 1000nM *Sc*Top2$^{\Delta CTD}$ were either nicked (to remove any remaining supercoils) or linearized and then separated by gel electrophoresis (**Figure 5D**). At 250nM *Sc*Top2$^{\Delta CTD}$, the major products were a nicked circle and a well-resolved ladder of bands visible below this species. As the protein concentration increased to 500 and 1000nM, the abundance of the nicked circle product decreased while bands in the lower part of the ladder became more pronounced (linearization of the products collapsed all species into a single band that co-migrated with the control linear band across all three samples). The pattern seen in the nicked samples is typical of that produced by DNA knotting reactions, in which each band corresponds to a different knot species with a characteristic number of crossovers or nodes (knots with more crossovers are more compact and hence migrate more quickly in the gel) (**Figure 1**; *Hsieh, 1983*; *Liu et al., 1980*; *Wasserman and Cozzarelli, 1991*). A repeat of the nicking experiment using full-length ScTop2 shows that this enzyme also forms knots, but that catenanes are again the principal product produced when condensation is permitted to occur (**Figure 5—figure supplement 1B**). Collectively, these data establish that the catalytic core of topo II is highly efficient at forming DNA knots at physiological enzyme concentrations, and that this activity can be routed into robust catenane production by the presence of the enzyme's CTD.

## Phase separation also occurs with human topo IIs to drive DNA catenation

Although poorly conserved in terms of amino acid sequence order (**Figure 2—figure supplement 1**), the general composition of human topo II CTDs – charge-rich, aromatic-poor, a general lack of folded structure – parallels that of the *Sc*Top2 CTD (**Figure 2B**). We therefore asked whether the two topo II isoforms encoded by humans, TOP2α and TOP2β, can also form condensates. Both enzymes were purified, N-terminally labeled with a Cy3 fluorophore, and imaged by confocal microscopy in the presence and absence of a Cy5-labeled 200bp DNA substrate (**Figure 6A**). TOP2α weakly formed spherical bodies on its own in the absence of DNA (at 1µM), whereas TOP2β produced larger puncta

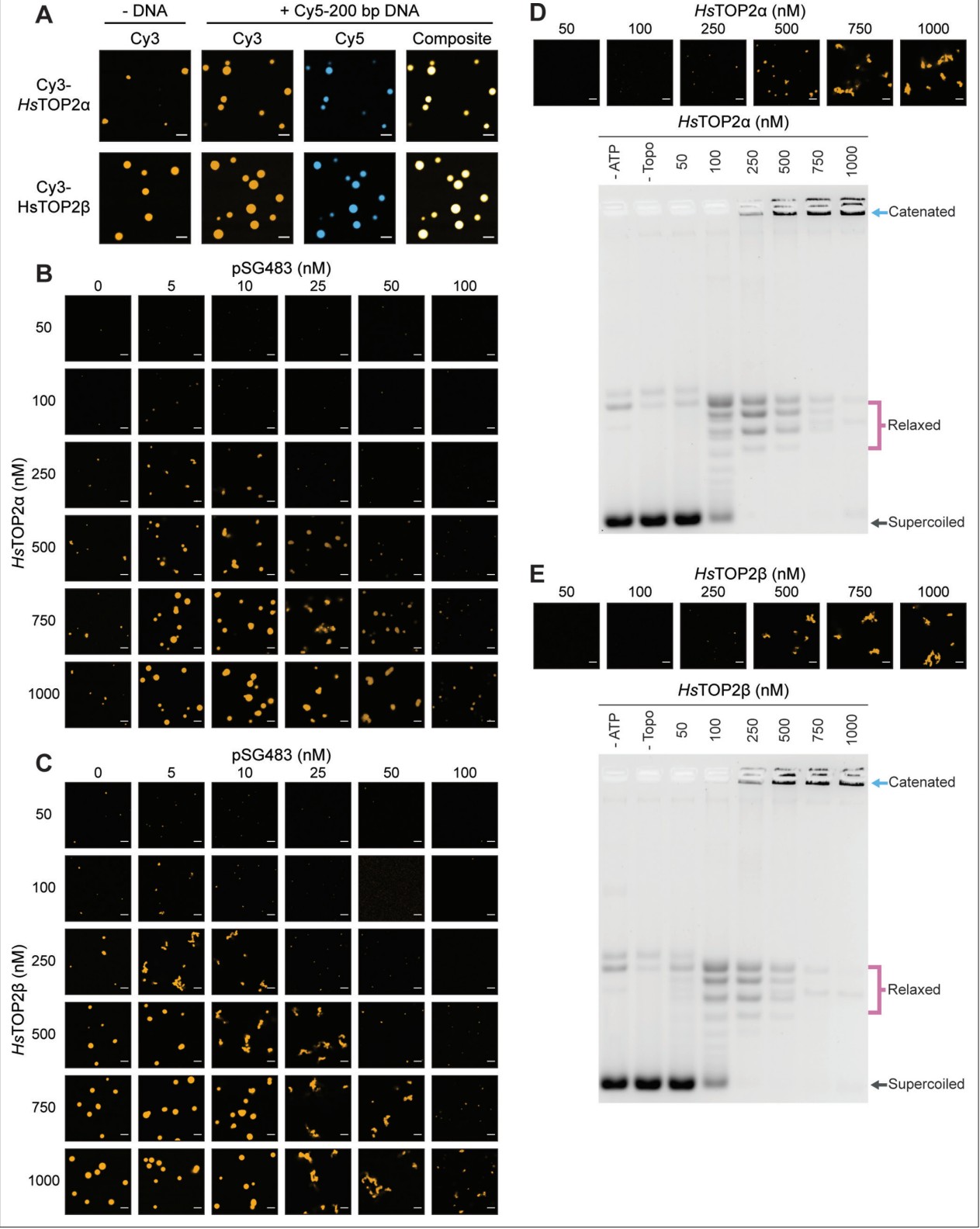

**Figure 6.** Human topo IIs form condensates and LLPS drives catenation. (**A**) Representative images of 1μM Cy3-*Hs*TOP2α and Cy3-*Hs*TOP2β each with and without 50nM Cy5-200 bp DNA. (**B, C**) Confocal images of a pairwise titration of different Cy3-*Hs*TOP2α and Cy3-*Hs*TOP2β concentrations, respectively, with different concentrations of pSG483. (**D, E**) Activity assays at different *Hs*TOP2α and *Hs*TOP2β concentrations, respectively, with 25nM

*Figure 6 continued*

pSG483. Concentration of topo II in the 'no ATP' lane is 1µM. Confocal micrographs of puncta were taken with Cy3-labeled protein and before adding ATP. Scale bars in (**A**) and (**B**) are 5µm.

The online version of this article includes the following figure supplement(s) for figure 6:

**Figure supplement 1.** Puncta formation by human topo IIs is stimulated by DNA.

under the same conditions. When 50nM of a Cy5-200 bp DNA substrate was added, TOP2α showed a marked increase in LLPS; by comparison, puncta formation by TOP2β was only moderately enhanced at this higher DNA concentration, but the labeled oligonucleotide colocalized with the two proteins in both instances. Thus, as with *Sc*Top2, human topo IIs also possess an ability to form condensates with DNA.

We next mapped out the phase response of TOP2α and TOP2β with respect to DNA and protein concentration using a plasmid substrate. TOP2α persistently showed a limited ability to form spherical puncta on its own, even at concentrations ≥250nM; these bodies were also substantially smaller than those of DNA-free *Sc*Top2 (≤3 µm² mean area) at the highest protein level tested (1µM) (*Figure 6B*, *Figure 6—figure supplement 1A*). By contrast, isolated TOP2β showed more spherical fluorescent bodies at lower protein concentrations (50–100nM), the size and number of which increased as protein concentration was increased until they reached an average dimension which exceeded that of DNA-free ScTop2 (*Figure 6C*, *Figure 6—figure supplement 1B*). For both proteins, the addition of pSG483 stimulated phase separation at low-to-moderate concentrations (more so for Top2α than Top2β) and resolubilized the condensates at higher levels. Collectively, these data show that while each of the eukaryotic TOP2s have slightly different condensate forming potential with respect to optimal protein and DNA concentration, they are all nonetheless capable of undergoing phase transitions in a manner stimulated by DNA (*Figure 6—figure supplement 1*. *Figure 7*). By contrast, isolated TOP2β showed more spherical fluorescent bodies at lower protein concentrations (50–100nM), the size and number of which increased as protein concentration was increased until they reached an average dimension which exceeded that of DNA-free ScTop2 (*Figure 6C*, *Figure 6—figure supplement 1B*). For both proteins, the addition of pSG483 stimulated phase separation at low-to-moderate concentrations (more so for Top2α than Top2β) and resolubilized the condensates at higher levels. Collectively, these data show that while each of the eukaryotic TOP2s have slightly different condensate forming potential with respect to optimal protein and DNA concentration, they are all nonetheless capable of undergoing phase transitions in a manner stimulated by DNA.

We next assessed the activities of TOP2α and TOP2β under conditions where condensates were either present or absent. TOP2α did not visibly begin to form condensates in the presence of 25nM plasmid DNA until reaching a protein concentration of 250nM. Accordingly, in the activity assay, DNA supercoil relaxation (but not catenation) was evident at 100nM enzyme; however, as TOP2α levels were increased to 250nM and beyond, puncta started to appear and catenated products became apparent and increased until all species were well-shifted (500–1000nM enzyme). An analysis of the activity of TOP2β as a function of LLPS showed a similar trend as with TOP2β, albeit with puncta and catenated species appearing at a slightly lower protein concentration (250nM), consistent with its enhanced condensation properties. Thus, the ability to couple catenation propensity to condensation status appears conserved across eukaryotic topo IIs.

## Discussion

During transcription, replication, and mitosis, discrete regions of DNA must be reversibly juxtaposed and separated in a highly controlled manner to support appropriate gene expression and cell division. Although type IIA topoisomerases regulate these topological transitions, how the enzymes selectively pass DNA strands to achieve desired outcomes (e.g. supercoil relaxation vs. decatenation) in an environment where duplex segments are densely packed is unknown. Here, we provide new insights into this question by showing that the catalytic core of eukaryotic topo II tends to form knots when acting on DNA at physiologic protein concentrations. We further show that at these same protein concentrations, the full-length protein – which bears an unstructured C-terminal element – efficiently forms phase-separated assemblies that locally concentrate DNA and promote DNA catenation.

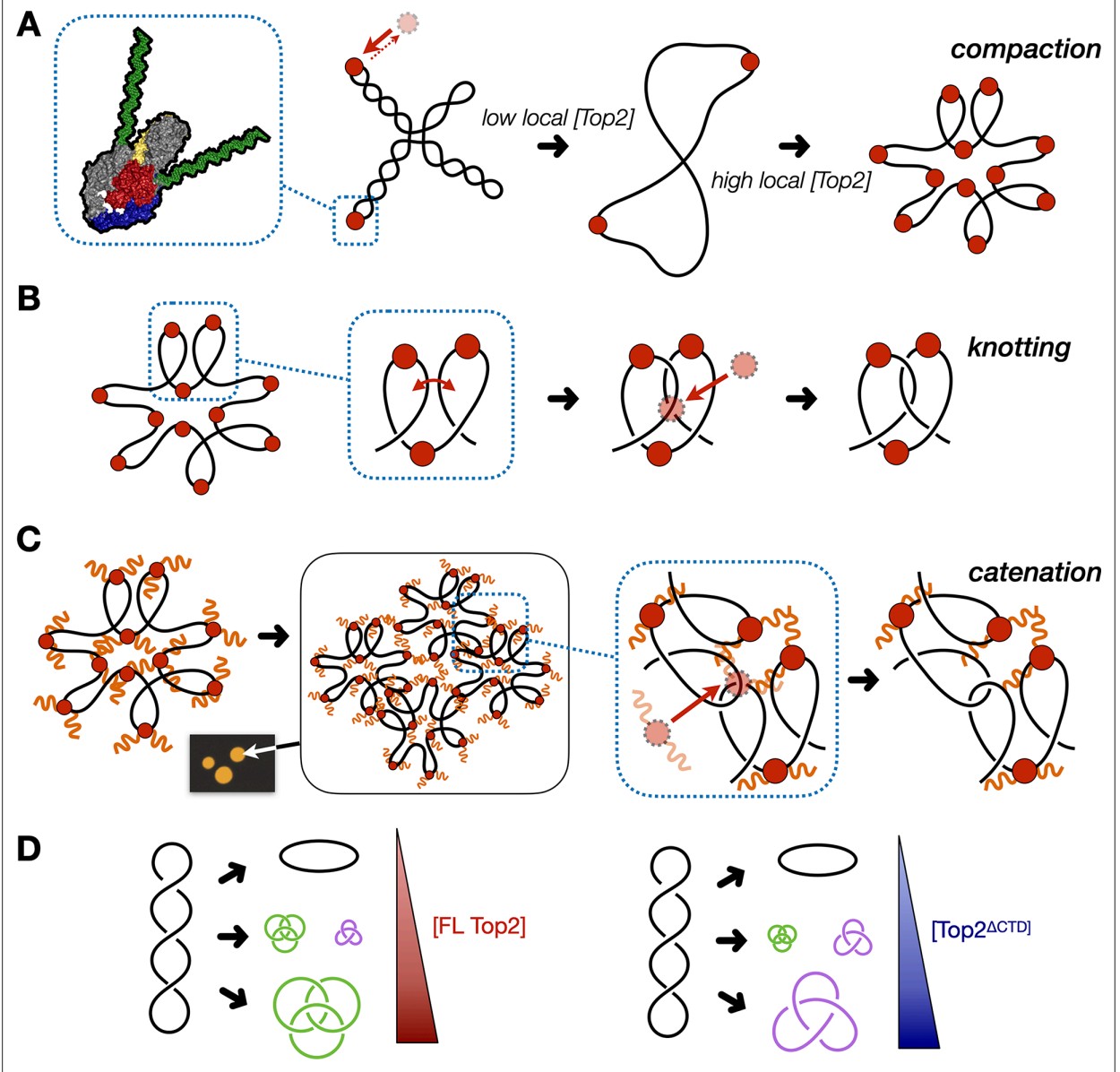

**Figure 7.** Model for multivalent interactions that drive topo II LLPS and how phase separation affects the steady state of topological states in vitro. (**A**) Low concentrations of topo II favor supercoil relaxation. Higher concentrations of topo II begin to compact DNA due to DNA bending by the cleavage core. (**B**) DNA compaction by topo II promotes knotting as crossovers transiently formed between loops within the same segment of DNA are acted upon on by the enzyme. (**C**) The topo II C-terminal domain aids in bridging together distal segments of DNA to favor the formation of inter-linked plasmids. (**D**) Schematic depicting how the relative distribution of topological species produced by topo II shifts as a function of protein concentration and depending on the presence or absence of the topo II CTD.

The online version of this article includes the following figure supplement(s) for figure 7:

**Figure supplement 1.** DNA binding with *Sc*Top2^ΔCTD contributes to knot formation.

## Protein-DNA interactions drive LLPS by ScTop2

Many proteins that undergo phase separation possess intrinsically disordered regions (IDRs) that facilitate multivalent interactions between themselves and other macromolecules to drive condensation. Consistent with this trend, we found that the C-terminal IDR of budding yeast topo II is necessary to promote LLPS by the enzyme (*Figure 4A*). Interestingly, the CTDs of eukaryotic topo IIs in general display relatively sparse sequence conservation between one another (*Figure 2—figure supplement 1*), but they nonetheless share certain chemical features such as a high fraction of charged residues and low aromatic amino-acid composition (*Figure 2B*). We found that the isolated *Sc*Top2 CTD was

unable to form condensates on its own but could do so in the presence of DNA or a crowding agent such as PEG (*Figure 4A*, *Figure 4—figure supplement 1*). We also found that higher salt levels blocked this phase transition. This behavior indicates that the topo II CTD can interact with itself and that DNA can serve as a scaffold to further increase local protein concentration through coacervation, a well-studied phenomenon by which charged polymers associate with each other to form condensates (*Aumiller and Keating, 2016*; *Banerjee et al., 2017*; *Pak et al., 2016*).

Full-length *Sc*Top2 also formed condensed bodies in the absence of DNA, as well as in the absence of PEG, at physiological salt (150mM KOAc) and protein concentrations (0.5–1.5µM protein, the reported concentration in human cells *Padget et al., 2000*; *Figure 2C*). These DNA-free assemblies did not recover after photobleaching, indicating that they correspond to a gel-like state, but they dissolved readily if salt levels were increased (200mM KOAc and above), demonstrating that they are not irreversible or insoluble aggregates (*Figure 4B*). Condensates were not visible when the *Sc*Top2 CTD was deleted, further indicating that this region can participate in protein-protein interactions to promote coalescence. Condensate-forming behavior seen in the absence of DNA may also be aided by dimerization of the topo II catalytic core, which generates a holoenzyme with two CTDs capable of facilitating multivalent interactions.

When DNA was present, the assemblies formed by full-length *Sc*Top2 became larger and fluid. These bodies in turn could be dissolved either by increasing salt to non-physiological levels (400mM) or by increasing nucleic acid concentration (*Figure 4B*). The difference in salt and DNA sensitivities between the isolated *Sc*Top2 CTD and the full-length enzyme is likely due to additional contacts with DNA that occur through the cleavage core of the enzyme (*Dong and Berger, 2007*) and the ATPase region (*Laponogov et al., 2018*). The type IIA topoisomerase cleavage core binds 20–30bp of DNA (*Dong and Berger, 2007*; *Lee et al., 1989*; *Peng and Marians, 1995*; *Thomsen et al., 1990*); given that larger DNA substrates are needed to promote LLPS (≥100bp) (*Figure 2E*), it is likely that DNA acts as a bridge to enhance condensation for native topo II as it does with the CTD alone. Collectively, our data indicate that multiple protein-DNA contacts, as well as protein-protein interactions between the CTDs, facilitate phase separation by *Sc*Top2. Defining precisely how the topo II CTD and its amino acid composition helps manifest LLPS at a molecular level will require further investigation by mutagenesis and biophysical study.

## The distribution of DNA products generated by topo II is controlled by local protein-DNA concentration and LLPS

It is well established that at low (1–10nM) protein concentrations in vitro, eukaryotic topo II will generate steady-state distributions of DNA products roughly in accord with thermodynamic expectations for the concentration of DNA used in the reaction (*Rybenkov et al., 1997*): that is, supercoiled DNA will be relaxed, while catenated and knotted DNAs will be unlinked. However, when superphysiological amounts of topo II are used in such experiments (5–10µM [*Padget et al., 2000*]), it has been shown that circular DNAs can now become catenated (*Hsieh, 1983*), even if the concentration of the nucleic acid substrate is maintained at a relatively low level (e.g., 25–75µM•bp, a value 100- to 1000-fold lower than present in a human nucleus). The mechanism underlying this switch in activity, which drives a topological outcome against what would otherwise be expected for the thermodynamic equilibrium, has been unclear. DNA catenation with low levels of topo II can be induced in vitro by the addition of proteinaceous (*Riou et al., 1985*; *Tse et al., 1984*) or chemical (*Krasnow and Cozzarelli, 1982*) DNA condensing agents, which increase the close proximity of DNA strands to favor linking; the ability of moderate concentrations of enzyme to do the same implies that topo II can directly condense DNA strands, a concept validated here. Interestingly, the C-terminal domain of topo II has proven necessary for observing catenation in both the presence (*Kawano et al., 2016*) and absence (*Shintomi and Hirano, 2021*) of exogenous DNA condensing agents. We now show that this element is directly responsible for facilitating DNA and enzyme coalescence.

Our biochemical studies show that condensates formed by *Sc*Top2 integrate DNA to promote catenation and that the dissolution of these condensates drives decatenation (*Figure 5B*). This behavior is shared by both human topo II isoforms, TOP2α and TOP2β, suggesting it is general to the eukaryotic type IIA topoisomerase family. Interestingly, removal of the *Sc*Top2 CTD not only disrupted LLPS, but also greatly impeded catenation and instead revealed robust knotting activity by the enzyme, even at sub-physiologic protein concentrations (250nM) (*Figure 5C and D*). Populations of DNA knots have

been previously reported when treating supercoiled plasmid with highly elevated levels of eukaryotic topo II (at≥20:1 enzyme:DNA molar ratios) (*Hirose et al., 1988*; *Hsieh, 1983*). DNA knots are also known to be created by the action of T4 bacteriophage topo II, which is composed of three subunits homologous to the ATPase domain and cleavage core of eukaryotic topo II but naturally lacks a C-terminal IDR (*Huang, 1986a*; *Huang, 1986b*; *Liu et al., 1979*; *Seasholtz and Greenberg, 1983*). Indeed, high concentrations of T4 topo II lead to the production of knot ladders (*Wasserman and Cozzarelli, 1991*) similar to what is observed here for *Sc*Top2 in the absence of its CTD (*Figure 5D*).

Why should elevated topo II levels lead to knotting and catenation? The Top2 core does not possess an inherent ability to distinguish crossovers within the same plasmid or between different plasmids. Despite this indiscriminate activity, a likely factor contributing to the ability of type II topoisomerases that lack an IDR to form knots, such as T4 topo II or *Sc*Top2$^{\Delta CTD}$, is the sharp DNA bend introduced by the cleavage region of the enzyme (*Dong and Berger, 2007*; *Vologodskii et al., 2001*; *Figure 7A*). At low protein concentrations, any given plasmid should have only a few enzymes bound at a time, which favors the binding and resolution of crossovers already present in the substrate, such as those arising from plectonemic supercoils (the affinity of topo II for supercoiled DNA is at least 10 fold greater than for relaxed substrates [*Osheroff, 1986*; *Osheroff, 1987*; *Zechiedrich and Osheroff, 1990*]). However, as the relative protein to DNA concentration increases, more DNA bends will be introduced by topo II binding, leading to increased compaction of the DNA and an increased likelihood of crossovers forming between different segments of the same plasmid (*Figure 7B*). When these crossovers are acted upon by another molecule of topo II, they will generate a knotted link. This model predicts that the density of knot crossovers should be proportional to the level of plasmid compaction facilitated by topo II, which is in turn dependent on the local relative ratio of topo II to DNA. Such a pattern is borne out by our data and of others (*Figure 5D*; *Hsieh, 1983*; *Wasserman and Cozzarelli, 1991*).

The formation of catenanes by full-length topo II is likely an extension of the DNA condensing behavior seen during LLPS, brought about by the ability of the C-terminal IDRs to create bridges between plasmid molecules that locally concentrate DNA and topo II. Here, DNA condensation will facilitate the formation of crossovers between two DNA molecules, such that the action of topo II will lead to plasmid interlinking and the formation of catenanes (*Figure 7C*). We note that CTD-less *Sc*Top2 initially formed some catenated products in addition to knots at intermediate concentrations (250–500nM), but that the level of these products was relatively low compared to full-length topo II and that only knots were formed at the highest concentration of the CTD-less ScTop2 tested here (*Figure 5D*). We attribute this difference in behavior to the relative ability of topo II to pass strands between DNA loops formed *within* a plasmid as compared to *between* plasmids: as protein concentration increases, DNA will become sufficiently compacted such that strand passage in cis (forming knots) outcompetes passage in trans (forming catenanes) unless some mitigating factor is present to favor the formation of inter-DNA links (e.g., increasing local DNA concentration through condensation mediated by the topo II CTD). We note that at intermediate protein concentrations where condensate formation is inefficient (full-length topo II) or blocked (as when the CTD is missing), mixtures of knots, catenanes, and relaxed topoisomers also can be observed (e.g., *Figure 5C*), reflecting that the decision to form or dissolve links reflects an equilibrium balance between local enzyme/DNA concentration and the relative access to *cis vs. trans* crossovers (*Figure 7D*); however, as protein concentrations further increase, the presence of the topo II IDR strongly shifts this equilibrium toward catenation (when starting from supercoiled DNA), whereas its absence favors knotting. The observation that both knotting and catenation, but not supercoil relaxation, are abolished by high salt comports with this schema (*Figure 4B*, *Figure 7—figure supplement 1*): elevated ionic strength serves to both prevent the CTD from facilitating LLPS and weakens the primary interactions of DNA with the topo II catalytic core, antagonizing loop formation, and shifting enzyme activity toward favoring the passage of pre-existing crossovers, such as those found in plectonemically supercoiled substrates.

## Implications of activity switching for topo II function in vivo

Although type II topoisomerases are known for their ability to dissolve topological problems such as supercoils, knots, and catenanes, there is growing evidence that the enzymes may also selectively bring distant DNA segments in proximity to one another in cells. For example, topo II has been proposed to directly modulate DNA condensation, as depletion of the protein leads to perturbations in chromosomal structure (*Bower et al., 2010*; *Gonzalez et al., 2011*; *Hirano and Mitchison, 1993*;

*Nielsen et al., 2020*; *Sakaguchi and Kikuchi, 2004*). Topo II has also been found to be enriched at centromeres of condensed mitotic chromosomes and has been proposed to play a role in creating or stabilizing catenanes between sister chromatids both within these elements and in other chromosomal loci prior to metaphase (*Edgerton et al., 2016*; *Floridia et al., 2000*; *Rattner et al., 1996*; *Tavormina et al., 2002*). More recently, work from the Hirano lab has shown that the *X. laevis* Top2α CTD helps guide the enzyme to mitotic chromosomes to promote DNA catenation and chromosomal maturation during mitosis (*Shintomi and Hirano, 2021*). Our observation that topo II uses its C-terminal IDR elements to promote self-association and form enriched regions of high local protein concentration that facilitate DNA entanglement provides a molecular rationale for this collection of findings.

Interestingly, in vivo fluorescence recovery after photobleaching experiments with human TOP2α have previously led to the suggestion that the enzyme could adopt a condensed liquid phase state when associated with mitotic chromosomes (*Antoniou-Kourounioti et al., 2019*; *Lane et al., 2013*; *Tavormina et al., 2002*). We now demonstrate that topo II can undergo LLPS in vitro, which we show to be dependent on the enzyme's C-terminal IDR. Moreover, both *Sc*Top2 (*Cardenas et al., 1992*) and TOP2α are known to be differentially phosphorylated over the course of the cell cycle (*Burden and Sullivan, 1994*; *Wells and Hickson, 1995*), and disruption of a hyper-phosphorylated region in the TOP2α CTD has been shown to slow recovery following bleaching in cells (*Antoniou-Kourounioti et al., 2019*). These data comport with our findings showing that hypophosphorylation of *Sc*Top2 substantially lowers the fluidity of its condensates (*Figure 4E&F*), suggesting that phosphorylation status could be used to fine-tune the dynamics of higher order structures formed by topo II in vivo. Many labs have shown that topo II is enriched along condensed chromosomal axes during different stages of the cell cycle (*Chu et al., 2020*; *Maeshima and Laemmli, 2003*; *Nielsen et al., 2020*; *Rattner et al., 1996*; *Warburton and Earnshaw, 1997*). Although additional studies will be needed to determine whether these localized regions of TOP2α possess condensate-like properties, our findings suggest that the presence of such structures is coupled to clustering (and potentially LLPS) by topo II, perhaps in concert with specific partner proteins.

During our studies, we found that the LLPS response to DNA and protein concentration was closely matched for *Sc*Top2 and human TOP2α, with both proteins showing a strong dependency on DNA. By comparison, TOP2β more readily formed condensates on its own and was less influenced by DNA at higher protein concentrations. This difference in behavior may in part be attributable to the isoelectric points of the topo II CTDs, which are basic for both *Sc*Top2 and TOP2α but acidic for TOP2β (*Figure 2B*), and the relative propensities of this region to engage DNA. In this regard, it is interesting to note that TOP2β is principally involved in transcriptional regulation. Many proteins that participate in transcription have been shown to undergo LLPS in vitro (*Boehning et al., 2018*; *Boija et al., 2018*; *Sabari et al., 2018*; *Zamudio et al., 2019*), a property that has been suggested to be of utility in forming condensed 'hubs' that are enriched for transcription factors and RNA polymerase associated proteins during transcriptional bursting (*Hnisz et al., 2017*; *Palacio and Taatjes, 2022*). TOP2β has been directly linked to the proper progression of certain transcriptional programs, such as the activation of estrogen response genes (*Ju et al., 2006*) and the expression of neuronal early-response genes (*Madabhushi et al., 2015*) previous studies have additionally reported that the estrogen receptor can integrate into and enhance mediator condensates in the presence of estrogen (*Boija et al., 2018*). Although the relevance of condensate formation to transcription in vivo is still under debate (*Bhat et al., 2021*; *Hnisz et al., 2017*), the ability of Top2β to undergo LLPS could allow the enzyme to associate with transcription factories that are also part of a locally coalesced body. Indeed, the capacity for topo II and DNA to form an LLPS state could be useful for bringing spatially distant DNA segments such as promoters and enhancer elements together into a common locus, a localizing function that has been proposed for eukaryotic topo IIs (*Pommier et al., 2022*).

Under condensate-forming conditions, each of the eukaryotic topo IIs tested here created physical interlinks between DNA segments that led to an increase of knots and catenanes. In vivo, topo IIs have been shown to facilitate the formation of knots (*Valdés et al., 2019*), and work from the Marko lab and others have shown that catenanes are present in mitotic chromosomes (*Bauer et al., 2012*; *Kawamura et al., 2010*; *Valdés et al., 2018*). Such topological intermediates – which are typically considered detrimental if allowed to persist – require either dissolution or some means to regulate their formation. In this regard, condensin has recently been reported to serve as a factor that can both promote catenation and antagonize knot formation by topo II in vivo (*Dyson et al., 2021*;

*Shintomi and Hirano, 2021*). We now show that topo IIs also have an intrinsic capacity to actively switch between DNA linking and unlinking modes. Catenation can be promoted by phase separation, whereas knotting occurs when local protein concentrations are high and decatenation predominates when protein concentrations are low or DNA binding is weakened. This behavior provides a potential mechanism by which topo IIs could alternate between producing distinct topological products in response to specific contextual needs. Future studies will be needed to probe the extent to which topo II manifests LLPS at a biophysical level, and to define the extent to which this property aids in partner protein selection and biological function in cells.

## Methods

**Key resources table**

| Reagent type (species) or resource | Designation | Source or reference | Identifiers | Additional information |
|---|---|---|---|---|
| Peptide, recombinant protein | *Sc*Top2 | This paper | | Purified from *S. cerevisiae* |
| Peptide, recombinant protein | *Sc*Top2$^{\Delta CTD}$ | This paper | | Purified from *E. coli*, 1–1177 |
| Peptide, Recombinant protein | *Sc*Top2$^{CTD}$ | This paper | | Purified from *S. cerevisiae*, 1178–1428 |
| Peptide, Recombinant protein | *Hs*TOP2α | This paper | | Purified from *S. cerevisiae* |
| Peptide, Recombinant protein | *Hs*TOP2β | This paper | | Purified from *S. cerevisiae* |
| Peptide, recombinant protein | BamHI-HF | NEB | Cat#R3136T | |
| Peptide, recombinant protein | Nb.BbvCI | NEB | Cat#R0631S | |
| Recombinant DNA reagent | 12URA-B (plasmid) | Addgene | Cat#48304 | Yeast Expression Vector |
| Recombinant DNA reagent | 12URA-C (plasmid) | Addgene | Cat#48305 | Yeast Expression Vector |
| Recombinant DNA reagent | 1B (plasmid) | This paper | | Modified version with Hisx14-SUMO tag. |
| Recombinant DNA reagent | pUC57 | Thermofisher Scientific | Cat##SD0171 | |
| Sequence-based reagent | 50 bp-F | IDT | Oligo for annealing with R | CATGCATACACGAGCTGCACAAACGA GAGTGCTTGAACTGGACCTCTAGT |
| Sequence-based reagent | 50 bp-R | IDT | Oligo for annealing with R | ACTAGAGGTCCAGTTCAAGCACTCTCG TTTGTGCAGCTCGTGTATGCATG |
| Sequence-based reagent | 100 bp-F | IDT | PCR Primer | AGTGTGATGGATATCTGCAGA |
| Sequence-based reagent | 100 bp-R | IDT | PCR Primer | TTTAAGCGGTGCTAGAGCTT |
| Sequence-based reagent | 200 bp-F | IDT | PCR Primer | GGGAAACCTGTCGTGCC |
| Sequence-based reagent | 200 bp-R | IDT | PCR Primer | ACATGTTCTTTCCTGCGTTATCCCC |
| Chemical compound, drug | Cyanine3 NHS-Ester | Lumiprobe | Cat#21020 | |
| Chemical compound, drug | Cyanine5 NHS-Ester | Lumiprobe | Cat#23020 | |
| Chemical compound, drug | Heparin Sodium | Fisher Scientific | Cat#BP2425 | |

## Cloning expression vectors

Full-length *Sc*Top2, *Hs*TOP2α, and *Hs*TOP2β expression plasmids were cloned by PCR amplifying appropriate cDNAs with an N-terminal HRV 3 C tag. PCR amplified *Sc*Top2 and *Hs*TOP2α cDNAs were inserted into the 12URA-B (Addgene #48304) yeast expression vector while *Hs*TOP2β was inserted into the 12URA-C (Addgene #48305) yeast expression vector using Ligation Independent

Cloning (LIC). Both plasmids contain a galactose-inducible promoter for the expression of an N-term His$_6$-TEV (12URA-B) or an N-term His$_6$-MBP-TEV (12URA-C) protein. The ScTop2$^{\Delta CTD}$ (1–1177) expression plasmid was cloned as above without the N-terminal HRV3C tag into the 12URA-B vector.

The ScTop2$^{CTD}$ (1178–1428) expression plasmid was cloned by PCR amplification from full-length ScTop2 12URA-B vector followed by insertion into a modified 1-B (Addgene #29653) E. coli expression vector. This modified vector has an IPTG-inducible promoter for the expression proteins with an N-terminal His$_{14}$-SUMO tag.

## Protein expression

Full-length Top2 expression plasmids and the ScTop2$^{\Delta CTD}$ expression plasmid were transformed into BCY123 cells and plated on homemade SM$^+$Ura$^-$ plates containing glucose. A total of 50 mL SM$^+$Ura$^-$ starter cultures containing glucose were inoculated from these colonies and grown to saturation at 30 °C. Starter cultures were diluted 1:20 into 1 L intermediate cultures in SM$^+$Ura$^-$ media supplemented with 2% sodium-DL-lactate and 1.5% glycerol and grown reach saturation at 30 °C. SM$^+$Ura$^-$ with lactate/glycerol cultures then diluted 1:10 in 1 L YP expression media supplemented with 2% sodium-DL-lactate and 1.5% glycerol and grown at 30 °C to an OD$_{600}$ of 1.3–1.5. Protein expression was induced by adding 2% w/v galactose for 6 hr at 30 °C. Cells were harvested by centrifugation and pellets were resuspended with 2 mL 1 mM EDTA, 250 mM NaCl per liter of culture. Resuspended yeast were frozen by dropwise addition into liquid nitrogen and cells were stored at –80 °C.

The ScTop2 CTD expression vector was transformed into homemade BL21-DE3-pRIL chemically competent cells and plated on LB +Kanamycin plates. 50 mL starter cultures in 2xYT +Kan were inoculated from single colonies and grown at 37 °C until saturation. Starter cultures were next diluted 1:100 in 1 L LB +Kan media and grown at 37 °C to an OD$_{600}$ of 0.5–0.6. Protein expression was induced by the addition of 0.5 mM IPTG and cultures were incubated 37 °C for 3 hr. Cells were harvested by centrifugation and pellets were washed with 20 mM Tris-HCl (pH 8.5), 500 mM KCl, 20 mM imidazole, and 10% glycerol. Washed pellets were flash frozen in liquid-nitrogen and stored at –80°C.

## Protein purification

For purification of full-length topo II and ScTop2$^{\Delta CTD}$ purification from yeast, pellets were lysed by cryogenic grinding using the 6870 Freezer Mill (SPEX SamplePrep). Crushed cells were resuspended in 20 mM Tris-HCl (pH 8.5), 500 mM KCl, 20 mM imidazole, 10% glycerol, 0.5 mM TCEP, protease inhibitors (leupeptin, pepstatin, PMSF). Lysates were clarified by centrifugation at 15,000 rpm for 45 min in a JA 25.50 rotor and the supernatant filtered using 1.1 μm syringe filters (Thermo Scientific #722–2000). Filtered supernatant was loaded onto a HisTrap HP column (Cytiva #17524802), washed with 40 CV of resuspension buffer, then washed with 5 CV of resuspension buffer containing 150 mM KCl. Protein was eluted from the HisTrap column using resuspension buffer (containing 150 mM KCl and 200 mM imidazole) and captured directly on a HiTrap SP HP column (GE #17-1152-01). Protein was eluted from the HisTrap SP column using a 150–500 mM KCl gradient solution in 20 mM Tris-HCl (pH 8.5) and 10% glycerol. After elution, protein was concentrated in an Amicon Ultra-15 30 kDA centrifugal filter units (Millipore #UFC9030) and incubated overnight at 4 °C with His-tagged PreScission protease (purified in house, for full-length Top2s) or TEV protease (QB3 Macro Lab, for ScTop2$^{\Delta CTD}$). After cleavage, the protein was separated from uncleaved material and proteases by running cleavage reactions through a HisTrap column and collecting the flow through. Following concentration, cleaved protein was further purified by gel filtration using a Sephacryl S-400 column (GE Healthcare) with one of two buffers: (1) non-labeled protein was purified by gel filtration using buffer containing 20 mM Tris pH 7.9, 500 mM KCl, 10% glycerol, and 0.5 mM TCEP (after elution, this material was concentrated, flash frozen, and stored at –80 °C), or (2) N-terminally labeled protein was passed over a gel filtration column in a buffer containing 50 mM HEPES-KOH (pH 7.0), 500 mM KCl, and 10% glycerol. Phosphatase-treated protein was purified as per the labeled protein, except that during the PreScission cleavage step, protein was treated overnight with Lambda Protein Phosphatase (NEB #P0753S) at 4 °C.

After elution from the sizing column, protein destined for dye labeling was concentrated in an Amicon Ultra-15 centrifugal filter units (30 kDA, Millipore #UFC9030). These samples were then mixed at a molar ratio of 1:1 with Cyanine3 NHS-Ester (Lumiprobe #21020) and incubated overnight at 4 °C. Free dye was removed from labeled protein by buffer exchanging labeling reactions three times using a Zeba Spin Desalting Column (7 kDa MWCO, Thermo Scientific #89890) into 50 mM Tris-HCl (pH

7.9), 500 mM KCl, 10% glycerol, and 0.5 mM TCEP. The protein was then concentrated, flash frozen, and stored at –80 °C. The dye-conjugated reactions had a labeling efficiency ranging from 70 to 90%. For microscopy experiments, labeled and unlabeled protein were mixed such that the total amount of labeled protein was 5–10%.

For purification of the *Sc*Top2 CTD, *E. coli* cell pellets containing expressed protein were resuspended in 20 mM Tris-HCl (pH 8.5), 500 mM KCl, 20 mM imidazole, 10% glycerol, 0.5 mM TCEP, and protease inhibitors (leupeptin, pepstatin, PMSF). After resuspension, samples were sonicated, and the lysate was clarified and filtered as per the full-length proteins. Clarified supernatant was loaded onto HisTrap FF crude column (GE Healthcare #17-5286-01) and then washed with 40 CV of resuspension buffer. Protein was eluted from the nickel column directly onto an ion exchange protocol as detailed above, except that a HiTrap Q HP column (GE Healthcare #29-0513-25) was used. After elution from the Q column, protein was concentrated and incubated overnight at 4 °C with His-tagged SUMO protease purified in-house. After incubating overnight, the SUMO protease and uncleaved protein was removed by running the sample through a HisTrap FF crude column. Cleaved protein was concentrated in an Amicon Ultra-15 centrifugal filter units (30 kDA, Millipore #UFC9030) and passed through two subsequent gel filtration steps using a: (1) Hiload 16/60 Superdex 75 column (Cytiva #28989333) and (2) and Sephacryl S-300 column (GE Healthcare). Buffers used for this sizing step and subsequent labeling reactions were followed as for full-length proteins. The labeling reactions and flash freezing conditions were followed as for full-length topo IIs.

## DNA substrate purification

For the 50 bp DNA substrate, reverse complement 50 bp oligos (5′-CATGCATACACGAGCTGCAC AAACGAGAGTGCTTGAACTGGACCTCTAGT-3′ ('top') and 5′-ACTAGAGGTCCAGTTCAAGCACTC TCGTTTGTGCAGCTCGTGTATGCATG-3′ ('bottom')) were ordered from IDT. The 50 bp duplex was annealed by mixing equimolar ratios of both oligos, heating the sample at 95 °C, and cooling to ambient temperature over several hours. Annealed duplex was buffer exchanged into 50 mM HEPES-KOH (pH 7.5), flash frozen in liquid nitrogen, and stored at –20 °C. For 100 bp and 200 bp DNAs, duplexes were PCR amplified from pUC57 (ThermoFisher Scientific #SD0171), concentrated, and passed over an S-400 gel filtration column (GE Healthcare) with 50 mM HEPES-KOH (pH 7.9), 50 mM potassium acetate, and 1 mM DTT. Both duplexes were buffer exchanged into 50 mM HEPES-KOH (pH 7.5), flash frozen, and stored at –20 °C.

Cy5-labeled 200 bp DNA duplex was PCR amplified with the same oligonucleotide primers as the non-labeled duplex except for the incorporation of an amino-modification at the C6 position of the 5′-cytosine of in the 'top' oligonucleotide. After PCR amplification, reactions were run over an S-400 gel filtration column and peak fractions were buffer exchanged into 0.1 M sodium bicarbonate (pH 8.5). DNA was mixed with an 8-fold molar excess of Cyanine5 NHS-Ester (Lumiprobe #23020) and incubated overnight at ambient temperature. Excess dye was removed by ethanol precipitating the DNA three subsequent times. Labeled DNA was resuspended in 50 mM HEPES-KOH (pH 7.5), flash frozen, and stored at –20 °C. Labeled and unlabeled DNAs were mixed so the total labeled material used in microscopy experiments was between 5–10%.

Supercoiled pSG483 was maxiprepped from XL1-Blue *E. coli* using NucleoBond Xtra Maxi columns (Macherey-Nagel) followed by isopropanol precipitation. The topology of supercoiled plasmid was assessed using native agarose gel electrophoresis. Plasmid DNA was diluted in 50 mM HEPES-KOH (pH 7.5), flash frozen, and stored at –20 °C.

## Confocal microscopy, FRAP, and analysis

Proteins were buffer exchanged using Zeba Spin Desalting Column (7 kDa MWCO, Thermo Scientific #89890) into 50 mM HEPES-KOH (pH 7.5) and 600 mM KOAc. Protein was then concentrated using an Amicon Ultra-15 centrifugal filter units (30 kDA, Millipore #UFC9030). Protein concentration and label efficiency were calculated by measuring appropriate absorbances using a Nanodrop One spectrophotometer. Phase separation was induced by lowering the salt concentration of the reaction to 150 mM KOAc across two 1:1 serial dilutions and then incubated at ambient temperature for 1.5 hr. Samples then were loaded onto Nunc Lab-Tek II 8-well Chambered Coverglass (Thermo Scientific #155360) for 3.5 min before imaging with a Zeiss AxioObserver equipped with a LSM800 confocal module and GaAsP detectors using a 40 x/1.30 PlanNeofluar oil objective. Cy3 images were taken with a 561

solid-state laser line and Cy5 images were taken with a 639 diode laser line. For condensate fusion experiments, time-lapses were taken of the samples after placing on coverslip.

All micrographs were subjected to rolling background subtraction using ImageJ (*Schindelin et al., 2012*). For puncta quantification and analysis, regions of interests (ROIs) were generated from background-subtracted micrographs by thresholding using the MaxEntropy algorithm followed by segmentation using watershed separation in FIJI (*Schindelin et al., 2012*). Puncta intensities were measured from these ROIs and the resultant data was analyzed in R studio (*R Studio Team, 2015*). ROI measurements were consolidated from three different micrographs for each condition; any puncta with an area smaller than 0.510 $\mu m^2$ was filtered out since puncta smaller than this may be from background. After filtering, the number of puncta was calculated and the geometric mean of the puncta area calculated (geometric means were chosen for calculation due to the logarithmic distribution of puncta sizes under certain conditions). These data was then used to construct quasirandom plots using ggplot2 (*Wickham, 2016*).

Fluorescence recovery after photo bleaching was performed using the imaging protocol and confocal microscope as detailed above. After placing samples on the coverslip, a 1.3 $\mu m^2$ region was photobleached to between 30% and 50% intensity in each puncta and fluorescence recovery was monitored by time-lapse imaging. Puncta intensities and recovery were analyzed in the same manner as above using ImageJ (*Schindelin et al., 2012*). All subsequent data analysis was conducted in R studio (*R Studio Team, 2015*). Intensity values of photobleached regions were analyzed by putting them through background subtraction followed by correcting the intensities for fluorescence loss due to photobleaching of images. Relative fluorescence intensities are shown to emphasize the mobile fraction of proteins for each condition. To compare mobile phase between phosphatase and control samples, statistical analysis using Two-way ANOVA with Bonferroni post-hoc analysis was conducted in GraphPad Prism between the different conditions tested. Plots were generated using ggplot2 (*Wickham, 2016*).

## Topoisomerase activity assays

Proteins were buffer exchanged into buffer made of 50 mM HEPES pH 7.5, 600 mM KOAc, and 4 mM magnesium acetate. Protein and pSG483 were mixed such that the final solution contained 50 mM HEPES pH 7.5, 150 mM KOAc, 1 mM MgOAc, and 25 nM pSG483. Reactions were allowed to incubate for 1.5 hr to allow for condensates to form. Microscopy images were taken at the 90 minute mark with Cy3-labeled protein. After 90 min, 1 mM ATP was added and reactions were allowed to proceed for 5 min at ambient temperature. Reactions were stopped by the addition of 20 mM EDTA, 1% SDS, and 200 ng/μL Proteinase K. After Proteinase K addition, samples were incubated at 55 °C for 1 hr. DNA Gel Loading Dye (Thermo Scientific #R0611) was then added to each sample and loaded onto a 1.4% non-denaturing agarose gel. Gels were run for 18–20 hr at 25 V, stained with ethidium bromide, and imaged on ChemiDoc Touch Imaging System (Bio-Rad #1708370).

Two different activity assays were performed, which differed only after the ATP addition step. For activity assays that utilized restriction enzymes, Cutsmart Buffer (NEB #B7204S) and BamHI-HF (NEB #R3136T, for linearization) or Nb.BbvCI (NEB #R0631S, for nicking) was added after a 5 min incubation period with ATP. Samples were then incubated at 37 °C for 1 hr, after which EDTA, SDS, and Proteinase K were added as per above. For the catenation/decatenation switch assay, the reaction buffer supplemented with ATP and salt were adjusted such that the final solution conditions were 50 mM HEPES-KOH (pH 7.5), 400 mM KOAc, 750 nM *Sc*Top2, 18.75 nM pSG483, and 1 mM ATP. Sample was stopped after 5 min with EDTA, SDS, and Proteinase K. Samples were then analyzed using native agarose gel electrophoresis as described above.

## Phosphorylation analysis by LC-MS/MS

Two samples of native (untreated) *Sc*Top2 and lambda phosphatase-treated *Sc*Top2 were separated by SDS-PAGE using an 8–20% gradient gel. The gel was Coomassie stained and bands corresponding to purified protein were excised and sent to the Taplin Mass Spectrometry facility (Harvard) for analysis. The following analysis pipeline was used at the facility to investigate the phosphorylation status. Excised gel bands were cut into approximately 1 $mm^3$ pieces. Samples were reduced with 1 mM DTT for 30 min at 60 °C and then alkylated with 5 mM iodoacetamide for 15 min in the dark at room temperature. Gel pieces were then subjected to a modified in-gel trypsin digestion procedure

(*Shevchenko et al., 1996*). Gel pieces were washed and dehydrated with acetonitrile for 10 min, followed by removal of acetonitrile. Pieces were then completely dried in a speed-vac. Gel pieces were rehydrated with 50 mM ammonium bicarbonate containing 12.5 ng/µl modified sequencing-grade trypsin (Promega, Madison, WI) at 4 °C. Samples were then placed in a 37 °C room overnight. Peptides were later extracted by removing the ammonium bicarbonate solution, followed by one wash with a solution containing 50% acetonitrile and 1% formic acid. The extracts were then dried in a speed-vac (~1 hr). Samples were then stored at 4 °C until analysis.

On the day of analysis, samples were reconstituted in 5–10 µL of HPLC solvent A (2.5% acetonitrile, 0.1% formic acid). A nano-scale reverse-phase HPLC capillary column was created by packing 2.6 µm C18 spherical silica beads into a fused silica capillary (100 µm inner diameter x~30 cm length) with a flame-drawn tip (*Peng and Gygi, 2001*). After equilibrating the column, each sample was loaded by a Famos auto sampler (LC Packings, San Francisco CA) onto the column. A gradient was formed and peptides were eluted with increasing concentrations of solvent B (97.5% acetonitrile, 0.1% formic acid).

As each peptide was eluted, they were subjected to electrospray ionization and then shunted into an LTQ Orbitrap Velos Pro ion-trap mass spectrometer (Thermo Fisher Scientific, San Jose, CA). Eluting peptides were detected, isolated, and fragmented to produce a tandem mass spectrum of specific fragment ions for each peptide. Peptide sequences (and hence protein identity) were determined by matching protein or translated nucleotide databases with the acquired fragmentation pattern by the software program, Sequest (ThermoFinnigan, San Jose, CA) (*Eng et al., 1994*). The modification of 79.9663 mass units to serine, threonine, and tyrosine was included in the database searches to determine phosphopeptides. Phosphorylation assignments were determined by the Ascore algorithm (*Beausoleil et al., 2006*). All databases include a reversed version of all the sequences and the data was filtered to between a 1 and 2% peptide false discovery rate. Phosphorylation sites were determined by identifying sites that were confidently assigned across both samples that were sent.

## Acknowledgements

The authors are grateful to the Taplin Mass Spectrometry Facility at the Harvard Medical School for conducting the mass spectrometry analysis. The authors are also grateful to Geraldine Seydoux, John Nitiss, and members of the Berger lab for helpful discussions and critical reading of the manuscript. This work has been supported by the NIGMS and the NCI (T32-GM7445-43, to JJ, and R35-CA263778, to JMB).

## Additional information

### Competing interests

James M Berger: Reviewing editor, *eLife*. The other authors declare that no competing interests exist.

### Funding

| Funder | Grant reference number | Author |
|---|---|---|
| National Institute of General Medical Sciences | T32-GM7445-43 | Joshua Jeong |
| National Cancer Institute | R35-CA263778 | James M Berger |

The funders had no role in study design, data collection and interpretation, or the decision to submit the work for publication.

### Author contributions

Joshua Jeong, Conceptualization, Data curation, Formal analysis, Supervision, Funding acquisition, Validation, Investigation, Visualization, Methodology, Writing - original draft, Writing - review and editing; Joyce H Lee, Conceptualization, Data curation, Formal analysis, Validation, Investigation, Visualization, Methodology, Writing - original draft, Writing - review and editing; Claudia C Carcamo, Conceptualization, Methodology, Writing - review and editing; Matthew W Parker, Conceptualization,

Writing - review and editing; James M Berger, Conceptualization, Formal analysis, Supervision, Funding acquisition, Investigation, Visualization, Writing - review and editing

### Author ORCIDs
Claudia C Carcamo ⓘ http://orcid.org/0000-0002-2646-188X
Matthew W Parker ⓘ http://orcid.org/0000-0002-7571-0010
James M Berger ⓘ http://orcid.org/0000-0003-0666-1240

### Decision letter and Author response
Decision letter https://doi.org/10.7554/eLife.81786.sa1
Author response https://doi.org/10.7554/eLife.81786.sa2

## Additional files

### Supplementary files
• MDAR checklist

### Data availability
Dataset information was uploaded to Dryad. DOI: https://doi.org/10.5061/dryad.z08kprrgc.

The following dataset was generated:

| Author(s) | Year | Dataset title | Dataset URL | Database and Identifier |
| --- | --- | --- | --- | --- |
| Berger JM, Jeong J, Lee J, Carcamo C, Parker M | 2022 | DNA-stimulated liquid-liquid phase separation by eukaryotic topoisomerase II modulates catalytic function | https://dx.doi.org/10.5061/dryad.z08kprrgc | Dryad Digital Repository, 10.5061/dryad.z08kprrgc |

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
