## [Editor Report]

Type II topoisomerases are essential players in virtually every aspect of genome organization and function of all organisms. The in vitro data presented here clearly demonstrate that eukaryotic type II topoisomerases phase separate under physiological conditions, forming liquid-liquid condensates, and that the outcomes of type topoisomerase II activity on DNA are altered in these condensates. The experiments and methods are sound, clearly described, and fully support the insightful and carefully formulated interpretation of the data. This work has broad implications for dissecting and delineating the myriad fundamental roles of this centrally important molecule.

---

## [Decision Letter]

**Decision letter after peer review:**

Thank you for submitting your article "DNA-Stimulated Liquid-Liquid Phase Separation by Eukaryotic Topoisomerase II Modulates Catalytic Function" for consideration by *eLife*. Your article has been reviewed by 3 peer reviewers, and the evaluation has been overseen by Maria Spies as a Reviewing Editor and Kevin Struhl as the Senior Editor. The following individual involved in review of your submission has agreed to reveal their identity: Nancy Kleckner (Reviewer #3).

Essential revisions:

All reviewers agreed that this is an interesting and important paper and that the results justify the conclusions. Reviewers 1 and 2, however, raised several points that need to be addressed by either additional explanations or data (if the authors chose so). Specifically, clarification of the data shown in Figure 5. Biological repeats and the resultant quantification of the data shown in panels A and C, along with critical discussions on this issue, would help further strengthen the key conclusion of the current manuscript.

*Reviewer #1 (Recommendations for the authors):*

Figure 3A: at pSG483 concentrations of 25 nm and above the morphology of the condensates changes dramatically and from figure 3B it appears that the fluidity or exchange also decreases. Slightly for these condensates at 25 nm, the highest concentration of plasmid measured in 3B. Since the conditions under which the dramatically different morphology form are used in subsequent experiments, e.g. Figure 5, it would be good to verify that the altered morphology does not result in an entirely different LLPS phase or behavior. It is an interesting question whether these structures are physiological or represent artificial aggregation. Based on the legend of Figure 4B increasing salt or DNA concentration in the buffer inhibits the formation of these structures. Have the authors assessed the reversibility of these structures by adding additional salt or DNA after first forming them in low salt conditions?

In a similar line of questioning, the distinct morphology occurs only with the supercoiled DNA substrate. Have the authors repeated these measurements with relaxed and/or linearized pSG483? The topology of the DNA could in principle alter the condensate behavior by altering the affinity of the enzymes to the substrate, and by altering the local concentration of DNA, which may alter binding of CTDs to DNA substrates in cis versus in trans. In yet another related question, it would be worth understanding how the presence of ATP alters the process of phase condensation. Not only would the ATP result in the relaxation of the supercoiled substrate, but the ATPase cycle could alter the conformations and affinity of the enzyme for the substrate DNA.

Figure 5 Supplement 1 B – The supertitle should be changed, since it indicates that the protein concentration is 1000 nm, whereas the protein concentration is increasing as indicated below the top title. Also can the authors more clearly delineate which lanes were nicked?

LLPS experiments were done at ambient temperature, does condensation occur at 37C?

scTop2 and hsTop2b form droplet like structures without the addition of DNA, and these are significantly smaller and less dynamic compared to condensates formed with DNA. Can the authors distinguish if these entities are liquid-liquid condensates or aggregates?

Also, did the authors check whether the protein preparations are free of contaminating DNA or tried the addition of a nuclease?

At certain protein and DNA concentrations spherical, droplet like structures are formed. Do these structures mature in time into other forms?

*Reviewer #2 (Recommendations for the authors):*

1) Line 186: The authors use potassium acetate buffers in most of the experiments reported here presumably because this buffer system is most suitable to access the propensity for topo II-mediated LLPS. It would be helpful for the readers to state whether they can see LLPS in other buffer systems that have been commonly used in the study of topo II enzymology (e.g., Tris- and NaCl^-^based buffer).

2) Line 213: The authors state that "budding yeast topo II can form higher-order spherical assemblies at protein concentrations greater than 250 nM…". What is the intracellular concentration of topo II in budding yeast cells? Of course, the effective, local concentration on chromatin and chromosomes would be different from the bulk concentrations in the cell, but such an estimate would help readers' understanding of to what extent the topo II concentrations used in the experiments are physiologically relevant.

3) Line 277: Although the observation that the addition of DNA makes structures formed by topo II more dynamic and fluid is potentially interesting, no clear discussion about why this happens is provided.

4) Line 327: The observation that 1,6-HD promotes the fibril aggregation of topo II is striking. Is this commonly observed when IDRs with a similar amino-acid composition to that of the topo II CTD (highly charged and poorly aromatic) are exposed to 1,6-HD?

5) Figure 4C-F: The set of experiments studying the effect of phosphatase treatments on the propensity for LLPS is weak. It is not clear to what extent the phosphorylation status of the topo II fractions purified from overexpressing cells reflects its physiological phosphorylation status. Treatment of the phosphatase-treated fractions with protein kinases known to phosphorylate topo II would help substantiate the authors' conclusion.

6) Figure 5: Topo II lacking the CTD forms catenanes rather than knots at intermediate concentrations (250-500 nM). This observation somehow contradicts the model shown in Figure 7. The authors' discussion regarding this point (line 623-) is not very convincing. This issue needs to be clarified. To do so, quantification of the results shown in Figures 5A and 5C, along with experimental repeats, would be required.

7) Figure 6: The observation that human topo IIa and topo IIb also undergo LLPS is an important addition to the current study, but unfortunately this part is very weak. It is not shown whether the observed LLPS depends on the CTD of each enzyme. The discussion about the difference between topo IIa and topo IIb (lines 671-692) is too speculative.

8) Figure 7: The model proposed here largely overlaps with that proposed by Shintomi and Hirano (2021). It would be fair to cite this recent work with a similar conclusion more explicitly.

9) Line 664-666: Citations here may not be appropriate: the Kleckner lab is not the first to show that topo II is enriched along chromosome axes. The authors may want to cite original papers such as those from the Laemmli and Earnshaw labs, especially, if it is the authors' policy to cite very original papers, like those on topo II enzymology.

10) Line 701-702: The discussion here by citing Dyson et al. (2021) is difficult to follow. The logic behind this discussion needs to be tightened.

---

## [Author Response]

Reviewer #1 (Recommendations for the authors):Figure 3A: at pSG483 concentrations of 25 nm and above the morphology of the condensates changes dramatically and from figure 3B it appears that the fluidity or exchange also decreases. Slightly for these condensates at 25 nm, the highest concentration of plasmid measured in 3B. Since the conditions under which the dramatically different morphology form are used in subsequent experiments, e.g. Figure 5, it would be good to verify that the altered morphology does not result in an entirely different LLPS phase or behavior. It is an interesting question whether these structures are physiological or represent artificial aggregation. Based on the legend of Figure 4B increasing salt or DNA concentration in the buffer inhibits the formation of these structures. Have the authors assessed the reversibility of these structures by adding additional salt or DNA after first forming them in low salt conditions?

We agree that studying different morphology present at 25 nM pSG483 is interesting. Our findings indicate that these fibrillary, non-circular structures are not aggregates, as they can be dissolved with high salt; this experiment was performed in the activity assay shown in Figure 5B. The altered morphology does not alter the LLPS-induced behavior of *Sc*Top2 switching to catenation, as shown in the activity assay Author response image 1: here, the activity assay performed in the paper was repeated with 5, 10, and 25 nM pSG483 at 1 µM and 2 µM *Sc*Top2. From this experiment, *Sc*Top2 catenates DNA at lower concentrations of DNA where spherical puncta are observed (Figure 3A). We have not included these data in the revised paper but can if the referee feels it important to do so.

**Author response image 1. sa2fig1:** 

In a similar line of questioning, the distinct morphology occurs only with the supercoiled DNA substrate. Have the authors repeated these measurements with relaxed and/or linearized pSG483? The topology of the DNA could in principle alter the condensate behavior by altering the affinity of the enzymes to the substrate, and by altering the local concentration of DNA, which may alter binding of CTDs to DNA substrates in cis versus in trans.

This is a good question that we did not address in the original paper. Per the request, shown in Author response image 2 are new microscopy images obtained for ScTop2 with supercoiled, nicked, and linear pSG483. The morphological change occurs with all three substrates. Combined with the data for the 200 bp substrate, these new findings suggest that the fibrillar morphology appears with longer DNA substrates. As noted in the paper, this shape change has also been reported for HP1 (https://doi.org/10.7554/*eLife*.64563). We have not included these data in the revised paper but can do so if the referee desires.

In yet another related question, it would be worth understanding how the presence of ATP alters the process of phase condensation. Not only would the ATP result in the relaxation of the supercoiled substrate, but the ATPase cycle could alter the conformations and affinity of the enzyme for the substrate DNA.

This is also a good question that we had not considered. In author response image 3 we show the effects of ATP on condensate morphology. Overall, nucleotide appears to have little impact, except perhaps at 1 µM, where the fibrils appear a touch thicker. As these data do not add significantly to or change the primary conclusions of the paper, we have not included them in the revision.

**Author response image 3. sa2fig3:** 

Figure 5 Supplement 1 B – The supertitle should be changed, since it indicates that the protein concentration is 1000 nm, whereas the protein concentration is increasing as indicated below the top title. Also can the authors more clearly delineate which lanes were nicked?

This request has now been addressed. The figure legend also has been modified to note that all lanes contain nicked DNA.

LLPS experiments were done at ambient temperature, does condensation occur at 37C?

Yes. Author response image 4 shows condensation occurs at 30 °C for *Sc*Top2 and at 37 °C TOP2α. The scale bar in each of the images is 5 µM. As these data do not add significantly to or change the primary conclusions of the paper, we have not included them in the revision.

**Author response image 4. sa2fig4:** 

scTop2 and hsTop2b form droplet like structures without the addition of DNA, and these are significantly smaller and less dynamic compared to condensates formed with DNA. Can the authors distinguish if these entities are liquid-liquid condensates or aggregates?

The puncta formed by DNA-free *Sc*Top2 and Top2b do not appear to be amorphous aggregates since the bodies they form are discrete/spherical and reversible with higher salt. However, FRAP analyses (Figure 3B, green plot for *Sc*Top2) indicate that they are not dynamic, suggesting that are that they are more gel-like (as opposed to liquid-phase) assemblies. We did not include FRAP data for Top2b (or Top2a) in the revised manuscript, as we intend to use these studies for a separate work focusing on these two enzymes.

Also, did the authors check whether the protein preparations are free of contaminating DNA or tried the addition of a nuclease?

Yes, we tested for DNA contamination by checking the 260/280 ratios when measuring protein concentration. The absence of nuclease contamination is also evident in the ‘ATP’ and ‘-Topo’ controls shown in our activity assays (Figures 5-6).

At certain protein and DNA concentrations spherical, droplet like structures are formed. Do these structures mature in time into other forms?

This is a good question that we checked. The spherical droplets maintain their shape up to 120 minutes post incubation. We have included these data in Figure 2—figure supplement 3.

Reviewer #2 (Recommendations for the authors):1) Line 186: The authors use potassium acetate buffers in most of the experiments reported here presumably because this buffer system is most suitable to access the propensity for topo II-mediated LLPS. It would be helpful for the readers to state whether they can see LLPS in other buffer systems that have been commonly used in the study of topo II enzymology (e.g., Tris- and NaCl^-^based buffer).

This is a question frequently asked of us. As noted in the paper, we avoid Na^+^ and Cl^-^ because these are non-physiological ions for intracellular proteins, and we have seen that the specific activity of many of the enzymes we study improves in K-Glu/OAc buffers. However, historically, the reviewer is correct to note that many topoisomerase studies still use Na^+^ and/or Cl^-^. We repeated the salt titration (Author response image 5) with 20 mM Tris-Cl pH 7.5 and varying concentrations of NaCl. Samples were incubated for the same amount of time as with the K-OAc buffer conditions and imaged in the same manner. As can be seen, *Sc*Top2 LLPS still occurs in the presence of NaCl and follows trends similar to our K-OAc measurements in terms of being stimulated by DNA, solubilized by high DNA concentrations, and inhibited at 400 mM salt. We have not included these data in the revised paper but can do so if the referee desires.

**Author response image 5. sa2fig5:** 

2) Line 213: The authors state that "budding yeast topo II can form higher-order spherical assemblies at protein concentrations greater than 250 nM…". What is the intracellular concentration of topo II in budding yeast cells? Of course, the effective, local concentration on chromatin and chromosomes would be different from the bulk concentrations in the cell, but such an estimate would help readers' understanding of to what extent the topo II concentrations used in the experiments are physiologically relevant.

This statement is based on an estimate of ~2 µM for the concentration of *Sc*Top2. We arrived at this value by taking mass spectrometry data compiled in the *Saccharomyces* Genome Database (SGD, https://www.yeastgenome.org/), where the median abundance appears to be roughly 4000 molecules/cell. According to a paper by Jorgensen *et al.* (https://doi.org/10.1091/mbc.e06-10-0973), the volume of an average yeast nuclei is 3 µM^3^, so this gives us an approximation of 2 µM *Sc*Top2. We note that when estimating the concentration of human topo IIs (using data from (Padget, Pearson, and Austin, 2000)).

3) Line 277: Although the observation that the addition of DNA makes structures formed by topo II more dynamic and fluid is potentially interesting, no clear discussion about why this happens is provided.

We agree with the referee that this is an interesting question; however, we do not have a good explanation for change in fluidity currently. It could be that, in the absence of DNA, interactions vis-à-vis the C-terminal domains somehow predominate to make the assemblies more gel-like, and that these contacts are weakened/interrupted by interactions with DNA. Alternatively, perhaps DNA simply acts as a spacer to keep topo IIs from packing too closely together. As part of our planned future studies, we intend to address such questions by using mutagenesis to investigate how changes to the amino acid composition and/or order of the C-terminal domain impact condensate behavior. However, given that our current ideas are speculative at best, we feel it best to avoid commenting on the behavior in the present manuscript.

4) Line 327: The observation that 1,6-HD promotes the fibril aggregation of topo II is striking. Is this commonly observed when IDRs with a similar amino-acid composition to that of the topo II CTD (highly charged and poorly aromatic) are exposed to 1,6-HD?

We believe the aggregation of Topo II with 1,6-HD is due to the Topo II core being denatured by 1,6-HD. This is supported by decreased activity of the enzyme in (Figure 4figure supplement 2B). When we explored this with the purified *Sc*Top2 CTD, we saw no effect on CTD phase separation from the addition 1,6-HD. As these data did not add significantly to or change the primary conclusions of the paper, we did not include them

5) Figure 4C-F: The set of experiments studying the effect of phosphatase treatments on the propensity for LLPS is weak. It is not clear to what extent the phosphorylation status of the topo II fractions purified from overexpressing cells reflects its physiological phosphorylation status. Treatment of the phosphatase-treated fractions with protein kinases known to phosphorylate topo II would help substantiate the authors' conclusion.

We agree that questions involving the phosphorylation status of purified proteins can be fraught with uncertainties, particularly when proteins are overexpressed regardless of cell cycle status. Since a large and variable number of post-translational marks have been reported for the C-terminal domain of topo IIs (e.g., see (Bedez et al., 2018; Cardenas, Dang, Glover, and Gasser, 1992)), we did not seek to make any particular connections between phosphorylation status *per se* as in might pertain in vivo. Rather, given that the topo II CTD *is* phosphorylated, we set out simply to test whether removing those phosphates might impact condensate fluidity (it did). This change is consistent with observations in other LLPS systems (such as eukaryotic replication initiators and Pgranule proteins (Wang et al., 2014) (https://doi.org/10.7554/*eLife*.04591)), where altering post-translational modifications also has been reported to regulate condensate properties or propensity. Unfortunately, there is no one kinase that is known to have a particularly dominant impact on the phosphorylation state of topo II, so re-phosphorylating the protein to test this effect as requested is not presently feasible.

6) Figure 5: Topo II lacking the CTD forms catenanes rather than knots at intermediate concentrations (250-500 nM). This observation somehow contradicts the model shown in Figure 7. The authors' discussion regarding this point (line 623-) is not very convincing. This issue needs to be clarified. To do so, quantification of the results shown in Figures 5A and 5C, along with experimental repeats, would be required.

We apologize for the confusion. We did not intend for the figure or text to imply that catenation never occurs in the absence the CTD or that knotting never happens when the CTD is present. Rather, our data indicate that the CTD strongly shifts the equilibrium between the two modes of product formation. We have clarified this point in the text (Lines 634-646) and by adding an additional figure panel to Figure 7.

Regarding quantification, while this is possible for single circle knots (which form discrete bands), this is not possible for catenanes, which only partially migrate into the gel and tend to show degrees of enrichment around the gel wells that vary considerably depending on gel thickness, comb size, running conditions, staining time, temperature, and the degree of catenation. In addition, there are many intermediate topological products formed in these reactions (e.g., catenated dimers or trimers of knots with varying numbers of nodes in each DNA circle) that give rise to a certain level of persistent smearing, even after nicking. As a consequence, attempts to measure the catenanes and knots in 5A and 5C will be misleading, as the final concentration of each product cannot be accurately boxed and scanned for densitometric analysis.

7) Figure 6: The observation that human topo IIa and topo IIb also undergo LLPS is an important addition to the current study, but unfortunately this part is very weak. It is not shown whether the observed LLPS depends on the CTD of each enzyme. The discussion about the difference between topo IIa and topo IIb (lines 671-692) is too speculative.

We agree that studying the effects of the CTD on LLPS by human Top2a and Top2b is important; however, repeating all the studies reported here for the yeast enzyme with both human proteins is beyond the scope of the present manuscript. We are currently working toward obtaining data for a second paper that will more thoroughly examine LLPS by Top2a and Top2b. Regarding the discussion concerning Top2a and Top2b, we felt it important to acknowledge previous models and observations by others that our work potentially helps to explain. We have attempted to tighten this part of the Discussion but respectfully prefer to leave much of the gist of the original work in place as a means of stimulating discussion in the field.

8) Figure 7: The model proposed here largely overlaps with that proposed by Shintomi and Hirano (2021). It would be fair to cite this recent work with a similar conclusion more explicitly.

Excellent point – this reference has been added to the Discussion section (Lines 656-658).

9) Line 664-666: Citations here may not be appropriate: the Kleckner lab is not the first to show that topo II is enriched along chromosome axes. The authors may want to cite original papers such as those from the Laemmli and Earnshaw labs, especially, if it is the authors' policy to cite very original papers, like those on topo II enzymology.

We agree and have added the suggested references (Lines 666-667).

10) Line 701-702: The discussion here by citing Dyson et al. (2021) is difficult to follow. The logic behind this discussion needs to be tightened.

We have attempted to improve the logic and flow of this portion of the Discussion as suggested.